



# Sizing Response of the Ultra-High Sensitivity Aerosol Size Spectrometer (UHSAS) and Laser Aerosol Spectrometer (LAS) to Changes in Submicron Aerosol Composition and Refractive Index

Richard H. Moore[1*], Elizabeth B. Wiggins[1,2], Adam T. Ahern[3,4], Stephen Zimmerman[1], Lauren Montgomery[1], Pedro Campuzano Jost[4,5], Claire E. Robinson[1,6], Luke D. Ziemba[1], Edward L. Winstead[1,6], Bruce E. Anderson[1], Charles A. Brock[3], Matthew D. Brown[1,6], Gao Chen[1], Ewan C. Crosbie[1,6], Hongyu Guo[4,5], Jose L. Jimenez[4,5], Carolyn E. Jordan[1,7], Ming Lyu[8], Benjamin A. Nault[4,5†], Nicholas E. Rothfuss[9], Kevin J. Sanchez[1,2], Melinda Schueneman[4,5], Taylor J. Shingler[1], Michael A. Shook[1], Kenneth L. Thornhill[1,6], Nicholas L. Wagner[3,4], and Jian Wang[9]

[1]NASA Langley Research Center, Hampton, VA, USA
[2]NASA Postdoctoral Program, Universities Space Research Association, Columbia, MD, USA
[3]NOAA Chemical Sciences Laboratory, Boulder, CO, USA
[4]Cooperative Institute for Research in Environmental Sciences (CIRES), University of Colorado, Boulder, CO USA
[5]Department of Chemistry, University of Colorado, Boulder, CO USA
[6]Science Systems and Applications, Inc., Hampton, VA, USA
[7]National Institute of Aerospace, Hampton, VA, USA
[8]University of Alberta, Edmonton, AB, Canada
[9]Washington University in St. Louis, St. Louis, MO, USA
[†]Now at Center for Aerosol and Cloud Chemistry, Aerodyne Research, Inc., Billerica, MA, USA

*Correspondence to*: Richard H. Moore (richard.h.moore@nasa.gov)

**Abstract.** We evaluate the sensitivity of the size calibrations of two commercially-available, high-resolution optical particle sizers to changes in aerosol composition and complex refractive index (RI). The Droplet Measurement Technologies Ultra-High Sensitivity Aerosol Size Spectrometer (UHSAS) and the TSI, Inc. Laser Aerosol Spectrometer (LAS) are two commonly used instruments for measuring the portion of the aerosol size distribution with diameters larger than nominally 60-90 nm. Both instruments illuminate particles with a laser and relate the single-particle light scattering intensity and count rate measured over a wide range of angles to the size-dependent particle concentration. While the optical block geometry and flow system are similar for each instrument, a significant difference between the two models is the laser wavelength (1054 nm for the UHSAS and 633 nm for the LAS) and intensity (about 100x higher for the UHSAS), which may affect the way each instrument sizes non-spherical or absorbing aerosols. Here, we challenge the UHSAS and LAS with laboratory-generated, mobility-size-classified aerosols of known chemical composition to quantify changes in the optical size response relative to that of ammonium sulphate (RI of 1.52 + 0i at 532 nm) and NIST-traceable polystyrene latex spheres (PSLs with RI of 1.59 + 0i at 589 nm). Aerosol inorganic salt species are chosen to cover the real refractive index range of 1.32 to 1.78, while chosen light-absorbing carbonaceous aerosols include fullerene soot, nigrosine dye, humic acid, and fulvic acid standards. The instrument response is generally in good agreement with the electrical mobility diameter. However, large undersizing deviations are observed for the low-refractive-index fluoride salts and the strongly absorbing nigrosine dye and fullerene soot particles. Polydisperse size distributions for both fresh and aged wildfire smoke aerosols from the recent Fire Influence on Regional to Global Environments Experiment – Air Quality (FIREX-AQ) and the Cloud, Aerosol, and Monsoon Processes Philippines Experiment (CAMP²EX) airborne campaigns show good agreement between both optical sizers and contemporaneous electrical mobility sizing and particle time-of-flight mass spectrometric measurements. We assess the instrument uncertainties by interpolating the laboratory response curves using previously-reported RIs and size distributions for multiple aerosol type classifications. These results suggest that, while the optical sizers may underperform for



strongly absorbing laboratory compounds and fresh tailpipe emissions measurements, sampling aerosols within the atmospherically-relevant range of refractive indices are likely to be sized to better than ±10-20% uncertainty over the submicron

aerosol size range when using instruments calibrated with ammonium sulphate.

## 1 Introduction

The size distribution and optical properties of atmospheric aerosols are important primary inputs to radiative transfer calculations of direct climatic effects. The optically-active particle size range (larger than ~50-80 nm diameters) also corresponds to the subset of particles that can act as cloud condensation nuclei and further impact clouds and climate indirectly. Consequently, most recent

field campaigns include measurements of the number size distribution and its higher moments from some combination of mobility, optical, and aerodynamic sizing. Of these techniques, modern optical particle sizers are ideal for covering the size range of interest with both high size resolution as well as the high time resolution needed for airborne observations. However, it is well known that optical particle sizer measurements depend not only on the particle geometric size, but also on the light source wavelength(s), scattering geometry of the instrument, and the particle morphology and complex refractive index (RI; $m = n + ki$). A particle

with an imaginary refractive index component ($k$) of zero does not absorb light and is purely scattering. Under some environmental conditions, this dependency can be exploited to derive the particle effective density, shape factor, and/or RI in addition to the size distribution by combining the optical sizer with instruments based on other particle sizing techniques (e.g., electrical mobility or aerodynamic particle sizing) using the so-called 'alignment method' (Hand and Kreidenweis, 2002). Zimmerman et al. (2015) attempted to apply this method using combined scanning electrical mobility particle classification followed by UHSAS and LAS

sizing of laboratory-generated aerosols of varying complex RI, but found limited dynamic range in particle size changes for purely scattering particles smaller than 550 nm diameter and significant undersizing for absorbing species. Moore et al. (2017) used a UHSAS to measure the size distribution of fresh aircraft engine soot, which required re-calibration of the UHSAS instrument size bins with size-classified Mini-CAST soot aerosols (using a differential mobility analyser, DMA) in order to shift the UHSAS soot particle size distributions larger to be consistent with those from a Scanning Mobility Particle Sizer. Recent work during the

Atmospheric Tomography Mission (ATom) and Observations of Aerosols Above Clouds and their Interactions (ORACLES) field campaigns suggests that the undersizing of the absorbing aerosol particles by the UHSAS may be due to particle vaporization by the high instrument laser power similar to that observed in the single-particle soot photometer, SP2 (Kupc et al., 2018;Howell et al., 2020). However, that explanation would not explain the similar undersizing reported by Zimmerman et al. (2015) for the LAS with much lower laser power.

Numerous estimates of RI for atmospherically-relevant aerosol particles at visible wavelengths have been reported in the scientific literature from both in situ and remote sensing observations. It is essential to distinguish between measurements made under dry conditions from those made under ambient (i.e., elevated relative humidity) conditions as most airborne in situ measurements fall into the former category, while most of the remote sensing measurements fall into the latter category. The RI of a hydrated aerosol particle is typically lower than its dry RI because of the contribution of the condensed water (real RI of 1.33), and the RI of the

aqueous solution has been shown to scale with the solute mass fraction (Tang and Munkelwitz, 1991, 1994). Here, we give a brief overview of some prior measurements of dry, submicron aerosol RIs since these are the conditions relevant for the typical UHSAS and LAS modes of operation.

Dry aerosol RIs measured for diverse air mass types over the United States during the NASA Studies of Emissions and Atmospheric Composition, Clouds, and Climate Coupling by Regional Surveys (SEAC[4]RS) field campaign show remarkably constant average

values for the real RI component at 532 nm wavelength near 1.52, where the interquartile range of observations varied between



1.49 and 1.55 for urban, marine, and biogenic air types as well as for both wildland fires and agricultural biomass burning plumes (Shingler et al., 2016;Aldhaif et al., 2018). Chamber studies of secondary organic aerosol (SOA) formation from biogenic precursors under low $NO_X$ conditions yield aerosols with real refractive indices near 1.44, while anthropogenic SOA exhibit a higher RI around 1.55 (Kim and Paulson, 2013). SOA aging appears to lead to small changes in the real RI whose direction depends

on the specific chemistry. For example, He et al. (2018) show that biogenic SOA formed in an oxidation flow reactor are non-absorbing between 400-650 nm wavelengths, and the real RI decreases from approximately 1.55 to 1.45 with increasing OH exposure. Meanwhile, Cappa et al. (2011) found that organic aerosol real RIs increased from near 1.47 to 1.52-1.54 with increasing OH oxidation lifetime for squalene and azelaic acid systems. Retrieved real RI values of 1.56-1.59 for wildland fire smoke particles were obtained via the alignment method during the Yosemite Aerosol Characterization Study (YACS) field campaign, which were

in reasonable agreement with the RI calculated as a volume-weighted mean from the simultaneously measured particle chemical composition (McMeeking et al., 2005). Laboratory measurements of biomass burning aerosols from multiple studies find a similar range of values for the real refractive index (1.54-1.67), while also placing constraints on the range of imaginary RIs of 0.002-0.22 for a diverse set of fuels and fire conditions (Mack et al., 2010;Hungershoefer et al., 2008;Sumlin et al., 2018). Shepherd et al. (2018) quantified the real RI for insoluble organic aerosol extracts for samples collected in London (urban), Antarctica (remote),

and air influenced by woodsmoke. They found that the aerosol real RI at 589 nm wavelength increased from remote (1.47) to urban (1.48-1.52) to woodsmoke (1.58). The low value of the real RI for organic aerosols sampled in Antarctica is consistent with Jurányi and Weller (2019), who derived a real RI of 1.44±0.08 via the alignment of aerosol size distributions obtained with an electrical mobility sizer and a LAS. In summary, past measurements of submicron atmospheric aerosols suggest that the real RI at visible wavelengths is close to that for ammonium sulphate and sodium chloride ($n \sim$ 1.52-1.54) with some suggestion that organic-rich

SOA and remote aerosols may exhibit slightly lower real RIs ($n \sim$ 1.45-1.52), while biomass burning particles may exhibit slightly higher real RIs ($n \sim$ 1.54-1.60). Inorganic aerosols and SOA tend to be non-absorbing at the longer visible to near-infrared wavelengths, while biomass burning particles are likely to be weakly absorbing near the emission source ($k < 0.1$) (Forrister et al., 2015). It is also known that biomass burning aerosols undergo rapid evolution in the atmosphere and become even less absorbing after only a few hours (Kleinman et al., 2020).

The goals of this study are to understand how changes in the aerosol complex RI may impact the optical sizing from the LAS and UHSAS and to look for evidence of size distribution biases for both laboratory-generated aerosols and fresh-to-aged biomass burning plumes undergoing rapid evolution in the atmosphere. Simple theoretical calculations are performed to contextualize the experimental results. This work should be of interest to current UHSAS or LAS users interested in constraining their measurement uncertainties as well as those seeking to understand the performance trade-offs of each instrument in sizing submicron aerosols in

both laboratory and ambient settings.

## 2 Methods

### 2.1 Instrument Descriptions

The TSI Laser Aerosol Spectrometer (LAS; Model 3340A; https://www.tsi.com/products/particle-sizers/particle-size-spectrometers/laser-aerosol-spectrometer-3340a/) uses a helium-neon gas laser (intracavity power ~1-10 watt) that operates in the

$TEM_{00}$ spatial mode at 633 nm wavelength with a $1/e^2$ intensity diameter of approximately 400 μm (TSI, 2015). The particle-laden air stream is drawn into the instrument and is focused by a 500-μm-diameter sample flow nozzle surrounded by a 760-μm-diameter sheath flow nozzle. As each particle traverses the laser, it scatters light that is collected by two pairs of wide-angle Mangin mirrors with one pair focusing the scattered light onto an avalanche photo diode (APD) and the other pair focusing the light onto a low-





gain PIN photodiode. The photocurrent signals from each detector are converted to voltages, which are then fed to four different

gain stages (G3 and G2 are the high gain and low gain for the APD, respectively; G1 and G0 are the high gain and low gain for the PIN photodiode, respectively). Peak hold circuits for each gain stage track the rise of the scattering intensity as the particle crosses the laser, and the peak signal is digitized for subsequent pulse height analysis. Each particle event is triggered from the G3 detector when the signal exceeds a user-specified threshold value, and the gain stage peak signals are then sampled successively downward from G3 to find the first gain stage that is not saturated. Finally, the instrument calibration is used to find the

corresponding size bin, and the counter for that bin is incremented. Manufacturer specifications report that the LAS can detect particles as small as 90 nm diameter with greater than 50% efficiency and as large as 7.5 μm diameter (TSI, 2015).

The optical and flow systems of the Droplet Measurement Technologies (DMT) Ultra-High Sensitivity Aerosol Size Spectrometer (UHSAS; Model G; https://www.dropletmeasurement.com/product/ultra-high-sensitivity-aerosol-spectrometer/) are largely similar to those of the LAS except for notable differences in the laser power and wavelength. The UHSAS uses a $Nd^{3+}$:YLF solid

state laser (intracavity power ~1000 watt) that operates in the $TEM_{00}$ spatial mode at 1054 nm wavelength with a $1/e^2$ intensity diameter of approximately 600 μm (DMT, 2006). Multiple papers have noted the similarity between the UHSAS laser characteristics and those of the laser used in the DMT SP2 to incandesce and vaporize black carbon-containing particles (Kupc et al., 2018;Howell et al., 2020). Consequently, the UHSAS might partially vaporize absorbing aerosol particles and/or their coatings, which would result in them being undersized. Manufacturer specifications report that the UHSAS can detect particles as small as

55 nm diameter with greater than 50% efficiency and as large as 1 μm diameter (DMT, 2006).

Individual particle-by-particle information such as peak signals are not recorded by either the UHSAS or LAS software, so an accurate calibration is essential for correctly assigning the particle scattering signals to their respective size bins. Two different types of calibration information are used by the data acquisition software and are stored in the instrument configuration files: a relative calibration that scales and smooths the transitions between adjacent gain stages (the so-called "stitching" procedure) and

an absolute calibration that quantitatively captures the overall shape of the detector intensity versus particle size relationship. The dependence of scattering intensity on particle diameter, $d_p$, depends on the particle size parameter, $x = \frac{\pi d_p}{\lambda}$, where λ is the laser wavelength. For $x \ll 1$, scattering intensity scales with $d_p^6$ (Rayleigh scattering), while for $x \gg 1$, scattering intensity scales with $d_p^2$ (geometric scattering). For $x \sim 1$, the Mie theory equations for the scattering intensity of a spherical particle do not follow a simple scaling relationship. Thus, the shape of the combined calibration curves for the UHSAS, and particularly the lower-

wavelength LAS, need to capture these three different regimes of Mie theory across six decades of scattering intensity and two decades of particle size (see Figure 4). In practice, multiple curve fits are combined across each regime using 4-5 calibration particle size points.

The intracavity laser power of both instruments is monitored continuously with a reference detector, which can be used to diagnose laser power drifts or, mostly commonly, contamination of the optical windows of the sealed optical block. Typically, the laser

reference voltage is between 1.0-2.8 V for the LAS, and lower reference voltages may change the absolute calibration conversion from the detector voltages to particle size. Changes in incident laser power may be less of a concern for the UHSAS with its longer laser wavelength, as more of the lower end of the particle diameter range is within the Rayleigh scattering reigme ($x \ll 1$) where particle size scales with the sixth-root of scattering intensity (DMT, 2006). However, potential calibration changes resulting from laser power changes prior to or after optical window cleaning have not been examined in the literature to date.




### 2.2 Mie Theory Calculations

The light scattering properties of submicron, homogeneous spherical particles are well described by Mie theory (Mie, 1908). Here, we calculate the size-resolved particle scattering and extinction efficiencies for particles of varying real and imaginary RI using the BHMIE code (Bohren and Huffman, 1998), as implemented in the Igor Pro programming language (WaveMetrics, Lake

Oswego, OR, USA). The code is similar to that available online at http://cires1.colorado.edu/jimenez-group/wiki/index.php/Analysis_Software#Mie_Code. Additional scattering and extinction calculations integrated across the entire phase function are performed using the MiePlot computer program (Laven, 2003), which is available online at http://www.philiplaven.com/mieplot.htm. Our goal is to understand how changes in RI are likely to affect the particle scattering signal detected by the optical particle sizers. We account for the complex scattering geometry of the UHSAS and LAS by

integrating the theoretical scattering phase function over the previously-reported UHSAS collection angles of 33-147° with a hole in the centre of this region (72.5-104.8°) where light is not sampled as described by Kupc et al. (2018). In addition, we assume that the particles are spherical and normalize their scattering cross-section to that for ammonium sulphate in Figure 1 in order to focus on the relative scattering intensity changes as compared to the instrument calibration aerosol (see Figure 4 for non-normalized, theoretical scattering cross-sections for ammonium sulphate and PSLs). Figure 1a,b suggest that both instruments would tend to

undersize non-absorbing, accumulation mode particles with real refractive indices less than that of the ammonium sulphate calibration standard of 1.52, while oversizing particles with larger real refractive indices. This trend weakens as the particle size approaches and exceeds the instrument laser wavelength. The influence of the longer-wavelength, UHSAS infrared laser becomes especially apparent here with the transition occurring near 600-700 nm diameters for the UHSAS versus near 400-500 nm diameters for the LAS. Figure 1c,d shows that an absorbing aerosol with a real RI of 1.7 and non-zero imaginary component would be

expected to be oversized toward the lower end of the particle size range and undersized at the upper end of the size range, which is consistent with a slice of the surface at $n = 1.7$ in panels a and b. Here, we choose to examine the variable $k$ for $n = 1.7$ as it is known that absorbing brown and black carbon have higher real RIs than ammonium sulphate and this value is in the middle of the range between reported values for biomass burning aerosols ($n = 1.55$-$1.65$) and black carbon (n ~ 1.85-1.95). It is also consistent with prior estimates for biomass burning organic aerosol species (Saleh et al., 2014). As the aerosol becomes more strongly

absorbing and the imaginary RI increases, Figure 1c,d indicates that the smallest particles are oversized even more significantly, while larger particles are undersized. The transition from oversizing to undersizing is expected to be more pronounced for the LAS than the UHSAS.

In addition to the theoretical calculations for the UHSAS optical geometry, we also examine the total scattering intensity integrated over all angles (see Supplementary Figures S1-S4), which show very similar trends for the accumulation mode size ratio. More

significant differences become apparent at larger sizes, when the aerosol scattering phase function is strongly biased in the forward direction.

The contribution of absorption and scattering to the overall particle light extinction is commonly parameterized as a single-scattering albedo (SSA), which is the ratio of the particle scattering coefficient to the extinction coefficient. Theoretical SSAs integrated across all scattering angles for size-resolved absorbing aerosol (real RI of 1.7 and variable imaginary RI) are shown in

Figure 2 for the laser wavelengths of the LAS (panel a) and UHSAS (panel b). First, it is instructive to examine the SSA and imaginary RI relationship for the visible LAS laser wavelength (Figure 2a), since this is near the visible wavelengths at which SSA data are commonly reported. The relationship between SSA and imaginary RI is fairly monotonic in the instrument sizing range. Past literature indicates that the SSA of atmospheric aerosols far from emissions sources is close to unity, while lower values near 0.80-0.9 have been observed for fresh biomass burning plumes (Kleinman et al., 2020;Selimovic et al., 2019;Eck et al., 2013).

Laboratory-generated combustion sources yield aerosol with even lower SSAs approaching 0.2-0.4 for fire lab measurements



(Pokhrel et al., 2016). Meanwhile, Kim et al. (2015) report SSAs below 0.1 for propane burner flame soot, which is comprised of black carbon (retrieved RI of 2+0.63i at 660 nm wavelength) and organic material (retrieved RI of (1.47±0.02)+(0+0.01)i at 660 nm wavelength). From Figure 2a, we might then regard particles with imaginary RI greater than 0.1 and SSAs smaller than about 0.6 as strongly absorbing aerosols, while non-zero imaginary RIs less than 0.05 and SSAs greater than about 0.8 are considered to be weakly absorbing aerosols. This categorization is consistent with the descriptors used in Table 1 for the laboratory experiments described in the next section.

Figure 2 also tells us that as particles get smaller in the Rayleigh regime, their absorption decreases more slowly than their scattering. Consequently, the SSAs for the UHSAS laser wavelength tend to be lower than for the LAS laser wavelength. This occurs even as an increase in the real RI would enhance the overall particle scattering and expected instrument response (Figure 1c,d).

## 2.3 Laboratory Experimental Setup

We challenge the UHSAS and LAS instruments using dry, size-classified particles of varying complex RI. A list of the investigated chemical species is given in Table 1. Each species is dissolved in 18 MΩ ultrapure Milli-Q water, and a medical nebulizer is used to introduce the particles into a filtered air stream. Particles in the sample flow are subsequently dried by a silica gel diffusion dryer, charged with a Po[210] radioactive bipolar charger, and size classified with a Scanning Electrical Mobility Sizer (SEMS; Brechtel Model 2002) system that is operated at constant voltage. The monodisperse output flow from the SEMS is combined with filtered makeup air and then split between the UHSAS, the LAS, and a condensation particle counter (CPC, TSI Model 3775). The amount of filtered makeup air is adjusted to ensure approximately 1 L min$^{-1}$ aerosol flow through the SEMS, while the sheath flow is set at either 10.0 L min$^{-1}$ for particle diameters less than approximately 550 nm or 5.0 L min$^{-1}$ for larger particles. This change in the sheath:aerosol flow ratio results in broader peaks for the larger particles, but it does not change the peak mode diameter observed by the optical sizers, which is obtained from a Gaussian curve fit to the most-prominent size distribution peak. We ignore any less-prominent, larger-diameter peaks resulting from the presence of multiply-charged particles. The SEMS differential mobility analyser (DMA) is optimized for larger particles up to and exceeding 1 μm in diameter at a reasonable 5:1 sheath:aerosol flow ratio.

The SEMS software solves the DMA transfer function in order to set the appropriate voltage for a given particle diameter and sheath flow. NIST-traceable polystyrene latex spheres (PSLs; ThermoFisher Scientific 3000 and 4000 series nanoparticles) are used to verify the transfer function calculations and proper operation of the system. For each PSL standard, the SEMS diameter is varied stepwise and the CPC number concentration is recorded, resulting in the peaks in the lower half of Figure 3a and in Figure 3b. Each peak is fit to a normal distribution, and the peak mobility mode diameters are compared to the reported PSL mode diameters in the upper half of Figure 3a with excellent agreement (slope of 0.996±0.003).

Having established the NIST size traceability of the SEMS mobility sizer, we then use the SEMS to calibrate the LAS and UHSAS optical sizers with classified ammonium sulphate aerosols (RI of 1.52 + 0i at 532 nm), consistent with recent best practices (Brock et al., 2011;Brock et al., 2016;Sawamura et al., 2017;Brock et al., 2019). The ammonium sulphate calibration curves for both the UHSAS and LAS are shown as red circles in Figure 4, where the optical sizer detector voltages of all gain stages are scaled to the G3 channel. Also shown as blue squares in Figure 4 are calibration curves using PSL particles (RI of 1.59 + 0i at 589 nm), which are frequently used to calibrate the optical sizers by the instrument manufacturers. Both calibrations are largely indistinguishable from each other up to about 200 nm in size, above which the curves then diverge markedly. Mie theory particle scattering cross-sections (shown as light, dotted lines on the right, ordinate axes in Figure 4) exhibit similar functional forms to the empirically-

...



derived calibration curves; although, the theoretical differences between the ammonium sulphate and PSL aerosols are smaller than those actually observed. Thick, black lines in Figure 4 are 6th-order monomial fits to the first few ammonium sulphate calibration points, since this is the expected functional dependence for Rayleigh scattering of particles. The calibration curves begin to meaningfully deviate from the 6th-order dependence around 300 nm and 500 nm for the LAS and UHSAS, respectively. This transition is consistent with the onset of the Mie scattering resonances that complicate the RI dependences shown in Figure 1a,b.

### 2.4 Airborne Field Measurements


In addition to the size-resolved laboratory measurements, we also examine the consistency of the LAS and UHSAS size distributions with other particle instruments deployed during two recent airborne field campaigns: the NASA/NOAA Fire Influence on Regional to Global Environments and Air Quality (FIREX-AQ) mission and the NASA Cloud, Aerosol, and Monsoon Processes Philippines Experiment (CAMP²EX). FIREX-AQ used the NASA DC-8 aircraft for targeted near- and far-field sampling of

multiple wildfire smoke plumes in the western United States during July-September, 2019, while the NASA P-3B aircraft encountered an aged smoke plume east of the Philippines during one of nineteen CAMP²EX flights carried out in August-October, 2019. Biomass burning smoke plumes are ideal for validating the optical sizers' response as the main aerosol number size distribution mode is typically centred around 200-300 nm, which is firmly in the overlap region of electrical mobility sizers, optical sizers, and particle time-of-flight mass spectrometers. In addition, size distribution noise is reduced by having good particle

counting statistics, and it is expected that the transition from "fresh" to "aged" smoke corresponds to a rapid transition in the particle optical properties (e.g., fresh plume SSAs of around 0.80-0.90 that increase toward unity for aged plumes) that allows us to examine potential sizing biases associated with changes in the aerosol complex RI of real-world atmospheric particles (Kleinman et al., 2020;Selimovic et al., 2019;Eck et al., 2013).

Aerosol size distribution instrumentation deployed during FIREX-AQ include a TSI Scanning Mobility Particle Sizer (SMPS) and

a pair of LASs operated by the NASA Langley Aerosol Research Group (LARGE), a UHSAS operated by the NOAA Chemical Sciences Laboratory Aerosol Optical Properties (AOP) Group, and an Aerodyne High-Resolution, Time-of-Flight Aerosol Mass Spectrometer (HR-ToF-AMS, hereafter abbreviated as AMS) operated by the Jimenez Research Group at CU-Boulder (Canagaratna et al., 2007;DeCarlo et al., 2006;Nault et al., 2018). The AOP UHSAS flow system is modified to maintain a constant volumetric sample flow as described by Kupc et al. (2018). For most of the FIREX-AQ flights, the UHSAS and one of the LARGE

LASs were operated behind thermal denuders heated to 250°C and 350°C, respectively, in order to characterize volatility impacts on the aerosol size distribution. However, the UHSAS thermaldenuder was bypassed for the 3 August 2019 research flight and a small portion of the 7 August 2019 research flight, both of which sampled the Williams Flats Fire smoke plume (estimated smoke ages of 0.5 to 7 hrs. for the fresh plumes as well as more aged plumes farther afield). Consequently, we examine these flights to directly compare the UHSAS size distributions with those observed by the un-denuded LARGE LAS. Meanwhile, the non-denuded

LAS was operated behind a monotube Nafion dryer (Permapure MD-700 series) to ensure that the aerosol did not contain any residual water. During FIREX-AQ, the UHSAS and LAS were frequently operated behind a common bridge diluter system to reduce particle concentrations to reasonable levels where particle coincidence effects on instrument counting or sizing would be minimal. In addition, the LAS instrument sample flow was sometimes reduced from 60 standard cm³ min⁻¹ to 30 standard cm³ min⁻¹ to further limit the instrument particle count rate for the most intense biomass burning plume penetrations. Both the FIREX-AQ

LAS and UHSAS are size-calibrated with ammonium sulphate particles, and the data are archived at 1 Hz.

SMPS data are archived at 60-second intervals representing the 45-second measurement voltage upscan and 15-second voltage downscan times. A challenge when interpreting the airborne SMPS size distributions on the DC-8 flying at airspeeds of 150-200



m s⁻¹ is that the observed size distribution reflects the concentrations of the smallest particles (~10-20 nm) captured near the beginning of the sampling interval and the concentrations of the largest particles (~200-250 nm) captured towards the end of the
sampling interval (45 seconds later in time and roughly 8 km away in distance). Consequently, we use the 1-second LAS measurements to assess the range of size distribution variability encountered over the SMPS data interval and focus our comparisons on the mode size, which is likely to be less variable than the localized particle number concentrations within each fire plume.

Aerosol mass size distribution data are obtained from the AMS during FIREX-AQ for a duration of 3-5 seconds each minute, using
enhanced particle Time-of-Flight mode, while the instrument operated in fast mass spectrometry (FMS) mode for the remaining time (Kimmel et al., 2011). This method of operation is used to maximize the amount of time spent in FMS mode in order to capture sub-plume-scale heterogeneity; however, it limits the number of overlapping size distributions available to compare with the optical sizers to, typically, one per plume intercept. Particle size information is obtained from the particle time-of-flight measurements as the particles accelerate into the reduced vacuum environment of the AMS, and particle mass concentration size
distributions are reported in terms of the particle vacuum aerodynamic diameter, $D_{va}$. The measured mass size distribution, $dM/d\log D_{va}$, is related to the number size distribution as

$$\frac{dM}{d\log D_{va}} = \frac{\pi}{6}\rho D_{va}^3 \left(\frac{dN}{d\log D_{va}}\right) \tag{1}$$

where $dN/d\log D_{va}$ is the number size distribution, and we assume that the particles are spherical (which is reasonable for thickly coated BB particles, see also Slowik et al. (2004)). The density, $\rho$, is calculated as a weighted average of the inorganic salts and organic species (Salcedo et al., 2006), with the organic aerosol density being determined from the elemental ratios (Guo et al., 2020;Kuwata et al., 2012). For the purposes of comparing the AMS size distributions with those from the LAS, UHSAS, and SMPS, we scale the x-axis $D_{va}$ linearly by the density $1/(\rho[\text{g cm}^{-3}])$ to compute the spherical-equivalent, geometric diameter, $D_p$
(DeCarlo et al., 2004). It should be noted that the AMS sampled from a different inlet than the particle sizers (as described in Brock et al. (2019) for a different mission), and that the lower residence times in the AMS inlet might have an impact on comparisons of highly volatile aerosol. Also, for low volatility species such as sulphate and more oxidized organic aerosol (MO-OOA) the vaporization time on the AMS can be comparable to the particle time-of-flight, resulting in a size shift towards larger diameters for these species (Canagaratna et al., 2007). While this is not a problem for fresh smoke, it can potentially impact the larger size
portion of the distributions in more aged smoke.

The CAMP²EX LAS size calibration was evaluated in the field using PSLs, and the PSL-based size bins are converted to ammonium sulphate equivalent sizes using a power law adjustment. During CAMP²Ex, number size distributions of particles with diameter ranging from ~ 10 to ~ 500 nm were measured using a Fast-integrating Mobility Spectrometer (FIMS) (Wang et al., 2018). The FIMS measures particle sizes based on electrical mobility as in a traditional scanning mobility particle sizer (SMPS).
As FIMS detects particles of different sizes simultaneously instead of sequentially as in traditional SMPS, it provides aerosol size distribution with a much higher time resolution at 1 Hz (Kulkarni and Wang, 2006;Wang et al., 2017). The relative humidity (RH) of the aerosol sample was reduced to below ~30% using a Nafion dryer before the sample being introduced into the FIMS. Therefore, the measured size distributions are for dry aerosol particles.


## 3 Results and Discussion

### 3.1 Sizing of Laboratory-Generated Challenge Aerosols

Figure 5 compares the monodisperse aerosol peak sizes measured by the LAS and UHSAS to the SEMS mobility size set points, where panels a-d are ratios to emphasize sizing differences and panels e-f are 1:1 plots to illustrate the size trends. As discussed in Section 2.3, both the UHSAS and LAS are calibrated using DMA-classified ammonium sulphate aerosols, and the consistency of this calibration is apparent from the red circles. Two different sets of ammonium sulphate calibrations were performed during the

study period, and it appears that the second UHSAS calibration slightly undersizes the largest diameters. Better agreement across the entire size range is seen for the other calibration. Here, we include both curves to demonstrate both the uncertainties resulting from Mie resonances as well as differences in the absolute calibration curves. Both sizers show size deviations greater than 10-20% for the fluoride salts (real RIs < 1.4) and for PSLs (real RI of 1.59), and these deviations are particularly apparent for particle diameters greater than 400-500 nm. The agreement between the SEMS and UHSAS sizing is generally found to be better than that

for the SEMS and the LAS. This is consistent with the lower onset of Mie resonances at 633 nm wavelength that are apparent in Figure 1 and that cause the flattening of the LAS calibration curve for particle sizes larger than the G2 gain stage limit of ~300 nm (Figure 4). In contrast, the UHSAS calibration curve doesn't really start to flatten out until ~500 nm, which is where the Mie resonances become only slightly visible in Figure 5a,b. It's also notable that the lithium bromide and chloride salt aerosols are undersized by both instruments – by only a few percent by the UHSAS and a more significant ~10% by the LAS. This response is

contrary to Mie theory expectations that predict that these particles should be oversized by both optical sizers (Figure 1a,b). Some uncertainty may stem from potential asphericity of the particles; although, we might expect that cubic, crystalline salt particles would behave similarly to sodium chloride, which is sized accurately by both the UHSAS and LAS. This result is consistent with Cai et al. (2008), who also found sodium chloride and doublet PSL particles to be correctly sized by the UHSAS despite their particle shape factors being greater than unity. Alternatively, the salts may form hydrates with residual water and a lower refractive

index than that reported by Li (1976) for the pure component salt.

The sizing performance of the LAS and UHSAS against the SEMS size set points worsens significantly for the laboratory-generated absorbing aerosol species, as shown in Figure 6. Here, the weakly absorbing fulvic and humic acid species are slightly oversized below about 400 nm with more significant oversizing observed at the larger particle diameters. It is interesting to note that these compounds closely follow the PSL sizing points, which makes sense given their similar real RIs. Meanwhile, the strongly absorbing

fullerene soot and nigrosine dye particles are significantly undersized by the UHSAS at almost all particle sizes and by the LAS above about 400 nm in diameter. The LAS sizing response to the nigrosine dye particles roughly follows the behaviour expected from Figure 1e, where the LAS initially oversizes them by almost 20% before significantly undersizing them.

The discontinuity in the UHSAS fullerene and nigrosine curves in Figure 6d,f is caused by a stitching error between the G1 and G2 detectors that results in a double peak size distribution, and we have intentionally included these data to highlight this effect.

Zimmerman et al. (2015) also made measurements of laboratory-generated fullerene soot and nigrosine dye particles of electrical mobility diameters from 100-600 nm and observed a smoothly varying response curve without a discontinuity for both instruments. Although not as prominent as for the UHSAS, the slight inflection points in the LAS fullerene soot data near 200 nm and 480 nm in Figure 6c,e reflect the same stitching issue. While such a stitching error is indiscernible or only noticeable as a slight dip for a polydisperse size distribution (see, e.g., the lowest-right panel of Figure 10), it may become much more noticeable (and

confounding) for monodisperse aerosol as shown by this example. This illustrates a challenge in the interpretation of combined mobility and UHSAS or LAS optical sizing measurements since the stitching error is only identifiable from measuring over the full particle size range. If only a single, or a few, discrete DMA sizes are selected then the double peak might be misinterpreted as an externally mixed aerosol.





It is worth paying particular attention to the relationship between PSL particles and DMA-classified ammonium sulphate particles since these compounds are most widely used for calibrating the UHSAS and LAS. Typically, manufacturer-calibrated instruments are returned from service with a PSL-based calibration, and it is often convenient to spot check the instrument size calibration in the field using PSLs since only a nebulizer is required. However, prior work has shown that most atmospheric submicron aerosols have a RI closer to ammonium sulphate (1.52+0i) than to PSLs (1.59+0i) (Shingler et al., 2016), so recent aerosol measurement campaigns have started converting the PSL-calibrated optical size bins post-mission, or preferably directly calibrating the internal instrument calibration curve using DMA-classified ammonium sulphate particles. Examples of these relationships are shown in Figure 7, including LAS and UHSAS data taken in 2020 after the NASA FIREX-AQ campaign as well as data for the UHSAS obtained during the 2012 NASA DC3 project (which were obtained for DMA-classified particles using a similar methodology to that described above). First, it is notable that the ammonium sulphate and PSL sizing below about 200 nm diameter are within a few percent of each other and the difference increases to a maximum around 500-800 nm diameters. Second, attempts to post-correct the sizing data may choose to use a power-law fit, which is monotonic and well-posed at large particle sizes, but which also fails to capture the Mie resonances at larger particle sizes. Attempts to fit the data to a high-order polynomial are able to capture the Mie resonances, but become unrealistic near 1000 nm, which is particularly problematic for the LAS. In addition, we note that both curves for the UHSAS sizing are roughly consistent with each other; although, there appears to be a ~5% offset at the lower size range that is likely due to differences in the PSL-based internal instrument calibration during these missions. This is important to keep in mind when applying a parameterized correction between PSL and ammonium sulphate size bin calibrations as small changes in the instrument absolute calibration curve would impact any post-mission instrument size correction. Finally, it may be worthwhile to apply a calibration curve to the LAS that accounts for expected differences in atmospheric aerosol RIs for submicron and supermicron aerosols. The good agreement between the optical sizing response for PSLs and the weakly-absorbing fulvic and humic acids shown in Figure 6e,f hint that the PSL size calibration standard is most appropriate for the supermicron LAS size range to accurately size what are thought to be weakly-absorbing, coarse mode dust aerosols (Froyd et al., 2019). Future work should examine the LAS response to dry-generated dust aerosols and sea salt particles to provide guidance on the appropriate calibration aerosols to use in the instrument supermicron size range.

### 3.2 Sizing of Fresh and Aged Wildfire Smoke Aerosols

The FIREX-AQ field campaign provides a unique opportunity to evaluate the LAS and UHSAS sizing performance for accumulation mode, biomass burning aerosols as they undergo rapid chemical transformation in the hours after emission. Figure 8 shows a typical FIREX-AQ sampling strategy for the 3 August 2019 Williams Flats Fire, where the NASA DC-8 aircraft horizontally profiled the smoke plume cross-section at constant altitude and at successive downwind distances from the fire. The average amount of time that it took for the aircraft to travel between adjacent legs was 20-25% that of the time it took the smoke to travel the same distance (Wiggins et al., 2020). Consequently, the sampling was not Lagrangian and differences in smoke particle amount and properties between the legs reflect the combination of changing fire emissions as well as longer downwind processing of the plume. The thick outlined portion of the flight track in Figure 8 corresponds to the timeseries data shown in Figure 9, where the smoke age in the top panel is coloured using the same scale as Figure 8. It is notable that the repeated transects from 23:10 to 23:25 have slightly different estimated smoke ages despite being at the same geographical position, which is indicative of the reproducibility of this rough smoke age estimate based on aircraft-measured wind speed. The lower panels of Figure 9 show the particle size distributions measured by the LAS, UHSAS, and SMPS. The UHSAS thermaldenuder was turned on from roughly 23:00 to just after 23:15, so this portion of the timeseries is not shown. Most of the other missing data periods for the UHSAS are



when the system sampled filtered air to complete a zeroing procedure. The smoke plume transects are clearly discernible from the background aerosol size distributions. All instruments show an increase in the aerosol mode size from less than 200 nm in the

closest transects to 200-300 nm in the farthest downwind transects.

Particle number size distributions for eight smoke plume transects and the upwind sampling leg are shown in Figure 10. Comparison times are selected to line up with the AMS size distributions, which were typically taken for a 3-4-second period every minute on the minute. Corresponding LAS size distributions for the freshest plumes show reasonable agreement with the AMS number size distributions; although, the reduced sampling statistics for the longer-aged plumes introduce some noise in the smaller

size bins when converting from dM/dlogDp to dN/dlogDp. The size distributions also agree well with those from the UHSAS and the SMPS in terms of the mode size. For the longer-aged plumes, the LAS shows a slightly larger and narrower distribution than the UHSAS and SMPS of about 10%; although, the upper size limit of the SMPS makes it hard to say that the mode is actually being captured. An additional challenge in this comparison is that the SMPS size distributions occur over a 60-second scan. This is not a problem for sampling outside of the plume (e.g., the upwind flight leg panel in Figure 10), but changes in the smoke plume

intensity over a minute may result in differences in the aerosol number concentrations. This is especially apparent for the plume transects at 0.5 hrs. and at 2.4 hrs. in Figure 10, where the SMPS distributions are higher than both the LAS and the AMS. Investigating the 1 Hz LAS size distributions in a short time interval surrounding the measurement point indicates significant concentration variability, and additional data are included for the LAS for these transects that show better agreement with the SMPS concentration magnitude and are within only 4-8 seconds of the other size distributions.

Airborne measurements of the Williams Flats Fire smoke on 7 August 2020 included a combination of sampling transects in the fresh plume as well as 'aged smoke' at a considerable distance downwind of the fire. Figure 11 presents aerosol number and mass size distributions for three sampling transects covering the dynamic range of observed particle size distributions. As the smoke plume is advected downwind, the LAS number mode size increases from 180 nm to 250 nm at smoke ages of 0.9 hrs. and 2.75 hrs., respectively. The LAS number mode size of the aged smoke is ~250 nm, which underscores the rapid evolution of the smoke

aerosols in the early hours after emission. LAS mass mode sizes are similar across all three size distributions at around 350 nm. The sizing of the SMPS, LAS, UHSAS, and AMS shown in Figure 11 are in reasonable agreement with each other across all plume ages, and it does not appear from either Figure 10 or Figure 11 that any one instrument is significantly or systematically biased relative to the others.

It is also worth discussing two important features that become apparent in the Figure 10 size distributions, which are the

discontinuities in the UHSAS size distributions at ~130 nm and at ~255 nm. These sizes correspond to the G3-G2 and G2-G1 transitions of the calibration curves (similar to those for the NASA UHSAS shown in Figure 4). Anomalous peaks due to these stitching errors are present for all UHSAS curves in Figure 10; although, the increased counting statistics noise in the size distribution for the upwind flight leg makes it difficult to see the individual peaks. Gain stage stitching errors are not obvious for the LAS size distributions; however, some distributions exhibit a sharp drop-off in concentration near 460-480 nm that is present

in both number and, more apparent, in the mass size distributions. This sharp transition is noticeable in both the Figure 9 LAS timeseries as well as the mass size distributions shown in Figure 11. The transition is not from a stitching error as it occurs within the LAS G1 gain stage that extends from ~330 nm to over 1000 nm. Rather, it occurs at the kink in the LAS calibration curve caused by the onset of Mie resonances (Figure 4), where the flattening of the calibration curve makes the LAS size response particularly susceptible to errors in the instrument absolute calibration. Such an error in the instrument absolute calibration would

be difficult to post-correct merely by shifting the size bins using a functional fit (e.g., power law, polynomial) similar to those in Figure 7. This motivates adjusting the instrument calibration parameters in the field based on mobility-classified particle data for an RI consistent with atmospherically-relevant particles (i.e., ammonium sulphate). It is worth reminding the reader of the caution



in the TSI LAS manual that "the relative stitching will never be perfect and the ability to zoom in on these transition regions can overemphasize the stitching errors" (TSI, 2015). This is particularly true for the monodisperse aerosol measurements in Figures 5

and 6, where the UHSAS G2-G1 stitching errors for nigrosine dye and fullerene soot curves might incorrectly be interpreted as a bimodal size distribution or change in aerosol refractive index. For the polydisperse size distributions exemplified by Figures 10 and 11, the stitching errors are imperceptible at best or exert a negligible effect on the overall size distribution mode at worst.

Aged smoke plumes encountered in southeast Asia during the CAMP$^2$EX field campaign provide another data set that is ideal for evaluating the LAS performance against the state-of-the-art FIMS electrical mobility sizer. Unlike the SMPS deployed for FIREX-

AQ, the FIMS makes measurements at the same 1 Hz time resolution as the LAS, while also covering the particle diameter range up to 600 nm. Thus, it should be able to capture the entirety of the biomass burning number size distribution, which is shown for select 16 September 2019 flight legs in Figure 12. Two flight segments are shown in Figure 12a that correspond to the timeseries panels b and d. Then a subset of each timeseries is used to compute the geometric mean (∗ one geometric standard deviation) size distributions for the most intense portions of the plume, which are shown in panels c and e. While both flight segments are impacted

by biomass burning emissions, the size distributions shown in panels b and c are also impacted by pollution outflow including possible volcanic emissions. These additional aerosol sources contributed to the prominent Aitken mode that is particularly evident in panel c. The white traces in Figure 12b,d are the count mean diameter (CMD) for the portion of LAS and FIMS size distributions above 100 nm, and the ratio of these diameters is shown as the red trace in panels b and d. The LAS CMD tends to be less noisy than that computed for the FIMS, and both diameters vary between 200-260 nm for the biomass burning plumes and are near 150

nm for the background air. While the size distributions measured by the LAS and FIMS both capture the accumulation mode size distributions well, the LAS CMD appears to be systematically biased high relative to the FIMS CMD by about 10%. It is not obvious, however, that this difference is outside of the expected instrument size uncertainty or due to a change in the aerosol RI since the offset is consistent for both the biomass burning and background aerosols.

### 3.3 Sizing Errors From Aerosol Composition and RI Changes

The results so far show that laboratory compounds with a wide range of RIs produce widely-varying optical sizing responses (particularly for the largest particle sizes), but that field campaign size distribution comparisons do not indicate obvious compositionally-dependent systematic optical sizing biases relative to electrical mobility and particle time of flight techniques. The likely explanation is that real-world aerosols, even those in relatively fresh wildfire smoke plumes, have size distribution modes towards the lower end of the optical sizers' range and exist as internally mixed aerosols with relatively narrow RI ranges.

This explanation is supported by summary statistics reported for the NASA SEAC$^4$RS airborne campaign, which examined a diverse range of aerosol types including agricultural and wildfire biomass burning, marine, urban, biogenic, free troposphere, and background conditions (Aldhaif et al., 2018;Shingler et al., 2016;Espinosa et al., 2019). Here, we use the type-averaged size distributions and RI ranges reported by these studies along with the laboratory sizing curves for select species (Figures 5 and 6) to quantify the expected sizing errors for each instrument due to compositionally-dependent RI changes. Similar analyses are

performed for the FIREX-AQ size distributions (Figures 10 and 11) using a conservatively wide range for n = 1.5-1.65, while k is assumed to be dominated by the organic aerosol component and is estimated following Saleh et al. (2014). Surprisingly, the k values for successive downwind smoke plume transects showed little variability (~0.03 and ~0.008 at 633 nm and 1054 wavelengths, respectively), while the size distribution mode shifted markedly toward larger diameters. Laboratory size ratios for sodium fluoride, ammonium sulphate, sodium chloride, and Suwannee River Fulvic Acid (SRFA) are linearly regressed against

their reported real RIs for each size bin, and the results are shown in Tables 2 and 3 for the LAS and UHSAS, respectively. This



approach only accounts for particle refractive index changes and neglects undersizing due to potential volatilization of strongly absorbing aerosols (e.g., nigrosine dye and fullerene soot) whose imaginary RIs are much larger than those expected for atmospherically-relevant aerosols. Sizing errors for particle diameters below 500 nm are generally better than ±10% with some higher values expected for the upper end of the plausible biomass burning RI range. The sizing errors generally increase with

increasing particle size, which is consistent with the behaviour shown in Figures 5 and 6. In addition, the sizing error ranges for the smallest particles tend to be skewed toward under sizing, while both under- and over-sizing errors occur for the largest particles. Incorporating the size distribution information from SEAC⁴RS and FIREX-AQ helps to further focus our error analysis on the instrument size range that is most relevant for atmospheric aerosols. We use the laboratory size ratio regressions for each size distribution bin to shift the size distributions larger or smaller to capture the range of realistic RIs. We then compute summary

statistics for each distribution including the CMD, volume mean diameter (VMD), total integrated particle surface area, and total integrated particle volume. Tables 4 and 5 report the differences in these summary statistics associated with RI-dependent LAS and UHSAS sizing errors, respectively. Both number and volume mean diameters are within ±10% for all air mass types with slightly more undersizing uncertainty than oversizing uncertainty. This behaviour is consistent with mode sizes reported by Shingler et al. (2016) for SEAC⁴RS and those shown in Figures 10 and 11 for FIREX-AQ, which tend to be toward the lower end

of the LAS and UHSAS size ranges (< 300 nm diameter), where the instrument sizing biases are minimized. It is important to also note that the range of real RIs used for the FIREX-AQ sizing error estimates in Tables 2-5 are likely to be overly broad, particularly at the high end of the range. Consequently, the large range of error estimates should be interpreted as conservatively large, upper and lower limits. In summary, the synthesis of the laboratory and atmospheric data show that the LAS and UHSAS instrument performance may not be as bad as suggested merely by the laboratory results alone. This is because real-world, atmospheric

aerosols tend toward Aitken and accumulation mode size distributions that become increasingly internally mixed over time. These size modes are within the optimal sizing range where both the LAS and UHSAS are less sensitive to RI changes. In addition, the processing of aerosols that transitions their mixing state to a more homogenous, internally mixed population also serves to narrow the range of RIs toward values characteristic of less-absorbing aerosols composed of organics, sulphate, and nitrate salts that roughly bound the range of real RIs reported by Aldhaif et al. (2018) and in Tables 2-5.

**4 Summary and Conclusions**

Modern optical particle sizers like the DMT UHSAS and TSI LAS are invaluable tools for studying the atmospheric aerosol size distribution with high time and size resolution, which makes them ideal for airborne measurements. Both instruments are optimized for capturing the aerosol accumulation mode through the use of focused lasers and wide-angle collection and focusing optics but differ in terms of the laser power and wavelength. Consequently, we expect from Mie theory that the instruments should size

aerosols of varying refractive index (RI) differently. Theoretical calculations suggest that non-absorbing particles with real RIs less than that of the calibrant (i.e., ammonium sulphate particles) would be undersized, while those with real RIs greater than that of the calibrant would be oversized. Empirical results from laboratory-generated aerosols show limited dynamic range in RI-dependent sizing of both instruments over the real RI range of 1.52-1.78 for particles smaller than 300-400 nm, but significant undersizing of fluoride salts with real RIs of 1.32-1.39 was found for both instruments.

Overall, the UHSAS tends to perform better for non-absorbing submicron aerosols with a tighter ratio of $D_{p,UHSAS}/D_{p,mob}$ around unity than was observed for the LAS. This strong performance for non-absorbing aerosols makes sense, because the UHSAS has a number of design advantages relative to the LAS including the longer laser wavelength, which yields a more monotonic instrument size response for accumulation mode aerosols up to about 500-600 nm. Meanwhile, the more significant LAS size



biases are consistent with the expected Mie resonances at particle diameters larger than 300-400 nm. An additional advantage of
the UHSAS design for submicron aerosols is the higher laser power and optimization of all four gain stages toward detecting
smaller particles and achieving a higher submicron size resolution. These features make the UHSAS an excellent choice for many
atmospheric measurement applications.

For absorbing aerosol particles, however, the LAS instrument appears to outperform the UHSAS in the laboratory tests for both
weakly and strongly absorbing aerosol particles up to about 400-500 nm in diameter. While both instruments tend to oversize the
weakly absorbing particles, the UHSAS bias is ~5-10% versus the LAS bias of ~0-5% below 300-350 nm diameters. For larger
diameters, the onset of Mie resonances yields even larger sizing biases. The most striking difference between the two instruments,
however, is seen for the strongly absorbing fullerene soot and nigrosine dye particles where the UHSAS significantly undersizes
the particles by more than 20% at all particle diameters greater than 80-150 nm. The LAS response to these strongly absorbing
particles is more consistent with theoretical expectations by oversizing the particles below 200-400 nm and then undersizing them.
The radical departure of the UHSAS size response from theoretical expectations for these compounds lends support to the
hypothesis previously suggested in the literature that the high-powered UHSAS laser interacts with the particle and alters its size
similar to the operating principle of the SP2 (Kupc et al., 2018;Howell et al., 2020).

In addition to the laboratory tests, we also examine the UHSAS and LAS size distribution measurements against other particle
sizing techniques used in recent NASA airborne field campaigns that studied biomass burning smoke plumes. The consistency
between the number size distributions is remarkably good with the distribution mode sizes generally agreeing to within 10% or
better. There is also no evidence to suggest systematic biases between the optical sizers and instruments based on particle time-of-
flight or electrical mobility techniques that would indicate RI-dependent sizing errors even across a wide range of smoke plume
ages and concentrations as well as background conditions.

These results confirm past recommendations in the literature to reference the calibration of the LAS and UHSAS submicron size
bins to mobility-classified ammonium sulphate aerosols. Future work should explore supermicron size response of the LAS, which
would be relevant for studies in environments with elevated dust or sea salt aerosol concentrations. Overall, the results suggest
that, while the optical sizers may underperform for absorbing laboratory compounds and fresh tailpipe emissions measurements,
sampling aerosols within the atmospherically-relevant range of refractive indices are likely to be sized to better than ±10-20%
uncertainty over the submicron aerosol size range when using instruments calibrated with ammonium sulphate. Propagating this
size uncertainty to the higher-order moments suggests that the error in derived aerosol volume is less than ~15-23% on the lower
end and less than 6-11% on the upper end of realistic RIs for most aerosol types. Uncertainties in derived volumes for biomass
burning aerosols may be even higher (up to ~30%) owing to their typically larger diameter accumulation mode size range.

## Author contributions

RHM designed the laboratory experiments and wrote the first draft of the manuscript. SZ and LM carried out the laboratory
measurements. RHM, EBW, ATA, PCJ, CER, ELW, CAB, HG, JLJ, ML, BAN, MS, TJS, KLT, and NLW made the FIREX-AQ
airborne measurements. LDZ, ELW, ECC, NER, MAS, and JW made the CAMP2EX airborne measurements. RHM and ATA
performed the Mie theory calculations. All authors contributed to the data interpretation and manuscript revisions.

## Data Availability

Laboratory data are included as an HDF5 file in the Supplementary Information. The FIREX-AQ and CAMP2EX field campaign
data are publicly available at the NASA Airborne Science Data for Atmospheric Composition Archive:





https://doi.org/10.5067/SUBORBITAL/FIREXAQ2019/DATA001 and
https://doi.org/10.5067/Suborbital/CAMP2EX2018/DATA001.

**Code Availability**

The MiePlot program used to carry out the Mie theory calculations is available with instructions and examples online at
http://www.philiplaven.com/mieplot.htm, while a version of the Igor Pro Mie code (which is based on based on the
FORTRAN77 code in Bohren and Huffman (1998) and was initially ported to Igor by C. Brock) is available online at
http://cires1.colorado.edu/jimenez-group/wiki/index.php/Analysis_Software#Mie_Code.

**Competing Interests**

The authors declare that they have no conflict of interest.

**Acknowledgements**

This work was funded by the NASA Atmospheric Composition Focus Area, specifically the Tropospheric Chemistry Program
managed by Dr. Barry Lefer and the Radiation Science Program managed by Dr. Hal Maring. Additional funding support provided
by a NASA New Investigator Award and the NASA NAAMES Earth Venture Suborbital mission. EBW and KJS are supported
by NASA Postdoctoral Program fellowships. We thank the FIREX-AQ project scientists Jim Crawford, Shuka Schwarz, Carsten
Warneke, and Jack Dibb, as well as the pilots and crew of the NASA DC-8. We thank the CAMP2EX project scientist Jeff Reid
as well as the pilots and crew of the NASA P-3. MS, HG, BN, PCJ and JLJ were supported by NASA grants 80NSSC19K0124
and 80NSSC18K0630.

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



**Tables:**


**Table 1: Summary of investigated chemical compounds and their refractive indices as reported in the literature.**

| Chemical Compound | Type | Refractive Index (Wavelength) | Reference |
|---|---|---|---|
| Sodium Fluoride | Non-absorbing | 1.32 + 0i (630-1050 nm) | Li (1976) |
| Lithium Fluoride | Non-absorbing | 1.39 + 0i (630-1050 nm) | Li (1976) |
| Ammonium Sulphate | Non-absorbing | 1.52 + 0i (532 nm) | Toon et al. (1976) |
| Sodium Chloride | Non-absorbing | 1.54 + 0i (630-1050 nm) | Li (1976) |
|  |  | 1.546+0.003i (532 nm) | Abo Riziq et al. (2007) |
| Polystyrene Latex Spheres (PSLs) | Non-absorbing | 1.57 + 0i (700 nm) | He et al. (2018) |
|  |  | 1.58 + 0.0004i (630 nm) | Ma et al. (2003) |
|  |  | 1.58 + 0.001i (1050 nm) | Ma et al. (2003) |
|  |  | 1.59 + 0i (589 nm) | Manufacturer (ThermoScientific) |
| Lithium Chloride | Non-absorbing | 1.66 + 0i (630-1050 nm) | Li (1976) |
| Lithium Bromide | Non-absorbing | 1.78 + 0i (630-1050 nm) | Li (1976) |
| Suwannee River Fulvic Acid | Weakly absorbing | 1.63 + 0.004i (630 nm) | Bluvshtein et al. (2016) |
|  |  | 1.55 + 0.03i (700 nm) | He et al. (2018) |
|  |  | 1.63 + 0.02i (532 nm) | Dinar et al. (2008) |
| Suwannee River Humic Acid | Weakly absorbing |  |  |
| Pahokee Peat Humic Acid | Weakly absorbing | 1.59+0.01i (630 nm) | Bluvshtein et al. (2016) |
|  |  | 1.56+0.03i (532 nm) | Michel Flores et al. (2012) |
| Fullerene Soot | Strongly absorbing | Estimated to be near 2 + 1i | Moteki and Kondo (2010) |
| Nigrosine Dye | Strongly absorbing | 1.67 + 0.26i (662 nm) | Garvey and Pinnick (1983) |
|  |  | 1.63 + 0.24i (532 nm) | Michel Flores et al. (2012) |




**Table 2: Expected percent LAS sizing error associated with an atmospherically-relevant range of RIs computed for each size by linearly interpolating between the laboratory response curves of NaF, (NH4)2SO4, NaCl, and SRFA as shown in Figures 5 and 6.**

| Aerosol Type | RI[a] | Expected LAS Sizing Error | | | |
|---|---|---|---|---|---|
| | | 100 nm | 250 nm | 500 nm | 800 nm |
| | | | | | |
| **SEAC4RS (Aldaif et al., 2018)** | | | | | |
| All | n = 1.47-1.57 | -5% to 1% | -8% to 0% | -7% to 7% | -9% to 9% |
| Agricultural Biomass Burning | n = 1.51-1.54 | -2% to 0% | -4% to -2% | -1% to 3% | -1% to 4% |
| Wildfire Biomass Burning | n = 1.50-1.57 | -3% to 1% | -6% to 0% | -3% to 7% | -4% to 9% |
| Background | n = 1.47-1.56 | -5% to 1% | -8% to 1% | -8% to 6% | -10% to 8% |
| Biogenic | n = 1.49-1.56 | -4% to 1% | -7% to -1% | -5% to 6% | -6% to 8% |
| Free Troposphere | n = 1.46-1.54 | -6% to -1% | -9% to -3% | -10% to 2% | -12% to 3% |
| Marine | n = 1.47-1.56 | -5% to 1% | -8% to -1% | -7% to 6% | -9% to 8% |
| Urban | n = 1.48-1.58 | -4% to 2% | -7% to 1% | -6% to 9% | -7% to 12% |
| Mix | n = 1.49-1.55 | -4% to 0% | -7% to -1% | -6% to 5% | -7% to 7% |
| | | | | | |
| **SEAC4RS (Espinosa et al., 2018)** | | | | | |
| Biogenic | n = 1.46-1.56 k = 0.002-0.006 | -5% to 1% | -9% to -1% | -9% to 6% | -11% to 8% |
| Biomass Burning | n = 1.50-1.60 k = 0.004-0.01 | -3% to 3% | -6% to 2% | -3% to 12% | -4% to 15% |
| Urban | n = 1.47-1.57 k = 0.003-0.007 | -5% to 1% | -8% to 0% | -8% to 7% | -10% to 9% |
| | | | | | |
| **FIREX-AQ (this study)** | | | | | |
| 03 August – Upwind Leg | n = 1.47-1.56 k = 0 | -5% to 1% | -8% to -1% | -8% to 6% | -10% to 8% |
| 03, 07 August – All Smoke ages | n = 1.5-1.65 k = 0.03 | -3% to 6% | -6% to 6% | -3% to 19% | -4% to 25% |

[a] RI range for SEAC4RS computed from literature values as the reported median ± 1.5 * the interquartile range for Aldaif et al.
(2018) and as the mean ± 2 * the standard deviation for Espinosa et al. (2018). For FIREX-AQ, the RI is assumed to be dominated by the organic aerosol component with a conservatively large range estimate for *n*, while *k* is computed following Saleh et al. (2014).






**Table 3: Expected percent UHSAS sizing error associated with an atmospherically-relevant range of RIs computed for each size by linearly interpolating between the laboratory response curves of NaF, (NH4)2SO4, NaCl, and SRFA as shown in Figures 5 and 6.**

| Aerosol Type | RI[a] | Expected UHSAS Sizing Error (%) | | | |
|---|---|---|---|---|---|
| | | 100 nm | 250 nm | 500 nm | 800 nm |
| | | | | | |
| **SEAC4RS (Aldaif et al., 2018)** | | | | | |
| All | n = 1.47-1.57 | -4% to 2% | -4% to 2% | -6% to 4% | -8% to 4% |
| Agricultural Biomass Burning | n = 1.51-1.54 | -2% to 0% | -1% to 1% | -2% to 1% | -3% to 1% |
| Wildfire Biomass Burning | n = 1.50-1.57 | -3% to 2% | -2% to 2% | -3% to 4% | -5% to 4% |
| Background | n = 1.47-1.56 | -5% to 1% | -4% to 2% | -6% to 3% | -9% to 3% |
| Biogenic | n = 1.49-1.56 | -3% to 1% | -3% to 2% | -4% to 3% | -7% to 3% |
| Free Troposphere | n = 1.46-1.54 | -5% to 0% | -5% to 0% | -8% to 1% | -11% to 0% |
| Marine | n = 1.47-1.56 | -4% to 1% | -4% to 2% | -6% to 3% | -8% to 3% |
| Urban | n = 1.48-1.58 | -4% to 3% | -3% to 1% | -5% to 2% | -7% to 2% |
| Mix | n = 1.49-1.55 | -4% to 1% | -3% to 1% | -5% to 2% | -7% to 2% |
| | | | | | |
| **SEAC4RS (Espinosa et al., 2018)** | | | | | |
| Biogenic | n = 1.46-1.56 k = 0.002-0.006 | -5% to 1% | -5% to 2% | -7% to 3% | -10% to 3% |
| Biomass Burning | n = 1.50-1.60 k = 0.004-0.01 | -3% to 4% | -2% to 5% | -3% to 7% | -5% to 8% |
| Urban | n = 1.47-1.57 k = 0.003-0.007 | -4% to 2% | -4% to 2% | -6% to 4% | -9% to 4% |
| | | | | | |
| **FIREX-AQ (this study)** | | | | | |
| 03 August – Upwind Leg | n = 1.47-1.56 k = 0 | -4% to 1% | -4% to 2% | -6% to 3% | -9% to 3% |
| 03, 07 August – All Smoke ages | n = 1.5-1.65 k = 0.008 | -3% to 7% | -2% to 8% | -3% to 12% | -5% to 14% |

[a] RI range for SEAC4RS computed from literature values as the reported median ± 1.5 * the interquartile range for Aldaif et al.
(2018) and as the mean ± 2 * the standard deviation for Espinosa et al. (2018). For FIREX-AQ, the RI is assumed to be dominated by the organic aerosol component with a conservatively large range estimate for *n*, while *k* is computed following Saleh et al. (2014).






**Table 4: Expected percent LAS sizing error associated with an atmospherically-relevant range of RIs computed for across characteristic size distributions by linearly interpolating between the laboratory response curves of NaF, (NH4)2SO4, NaCl, and SRFA as shown in 760 Figures 5 and 6. Size distributions for SEAC4RS are from Shingler et al. (2016), while FIREX-AQ size distributions are those shown for the LAS in Figures 10 and 11.**

| Aerosol Type | RI[a] | Expected LAS Sizing Error (%) | | | |
| | | Count Mean Diameter | Volume Mean Diameter | Total Surface Area | Total Volume |
| | | | | | |
| **SEAC4RS (Aldaif et al., 2018)** | | | | | |
| Agricultural Biomass Burning | n = 1.51-1.54 | -4% to -2% | -4% to -2% | -8% to -4% | -12% to -6% |
| Wildfire Biomass Burning | n = 1.50-1.57 | -5% to 0% | -5% to 1% | -10% to 1% | -15% to 3% |
| Background | n = 1.47-1.56 | -7% to 0% | -7% to 0% | -14% to -1% | -21% to -1% |
| Biogenic | n = 1.49-1.56 | -5% to -1% | -6% to 0% | -11% to -1% | -17% to 0% |
| Free Troposphere | n = 1.46-1.54 | -8% to -2% | -8% to -2% | -15% to -4% | -23% to -4% |
| Marine | n = 1.47-1.56 | -6% to -1% | -7% to -1% | -13% to -2% | -21% to -3% |
| Urban | n = 1.48-1.58 | -6% to 1% | -7% to 2% | -13% to 2% | -19% to 5% |
| | | | | | |
| **SEAC4RS (Espinosa et al., 2018)** | | | | | |
| Biogenic | n = 1.46-1.56 k = 0.002-0.006 | -8% to -1% | -8% to 0% | -15% to -1% | -23% to 0% |
| Biomass Burning | n = 1.50-1.60 k = 0.004-0.01 | -5% to 2% | -5% to 3% | -10% to 6% | -15% to 11% |
| Urban | n = 1.47-1.57 k = 0.003-0.007 | -7% to 0% | -8% to 1% | -14% to 1% | -21% to 3% |
| | | | | | |
| **FIREX-AQ (this study)** | | | | | |
| 03 August – Upwind Leg | n = 1.47-1.56 k = 0 | -7% to 0% | -8% to 1% | -14% to 1% | -21% to 4% |
| 03 August – 0.5 hrs. Smoke age | n = 1.5-1.65 k = 0.03 | -5% to 6% | -5% to 8% | -10% to 14% | -15% to 26% |
| 03 August – 1.2 hrs. Smoke age | n = 1.5-1.65 k = 0.03 | -5% to 6% | -6% to 8% | -10% to 15% | -16% to 26% |
| 03 August – 2.4 hrs. Smoke Age | n = 1.5-1.65 k = 0.03 | -5% to 7% | -5% to 9% | -10% to 17% | -14% to 31% |
| 03 August – 3.1 hrs. Smoke Age | n = 1.5-1.65 k = 0.03 | -5% to 7% | -5% to 9% | -10% to 17% | -14% to 31% |
| 03 August – 4.3 hrs. Smoke Age | n = 1.5-1.65 k = 0.03 | -5% to 8% | -5% to 10% | -10% to 18% | -14% to 33% |
| 03 August – 4.9 hrs. Smoke Age | n = 1.5-1.65 k = 0.03 | -5% to 8% | -5% to 10% | -10% to 19% | -15% to 32% |
| 03 August – 6.1 hrs. Smoke Age | n = 1.5-1.65 k = 0.03 | -5% to 8% | -5% to 10% | -10% to 18% | -15% to 31% |
| 03 August – 7.1 hrs. Smoke Age | n = 1.5-1.65 k = 0.03 | -5% to 8% | -5% to 10% | -10% to 19% | -14% to 33% |
| 07 August – 0.9 hrs. Smoke Age | n = 1.5-1.65 k = 0.03 | -5% to 7% | -6% to 9% | -10% to 16% | -16% to 29% |
| 07 August – 2.75 hrs. Smoke Age | n = 1.5-1.65 k = 0.03 | -5% to 8% | -5% to 10% | -10% to 18% | -14% to 33% |
| 07 August – Aged Smoke | n = 1.5-1.65 k = 0.03 | -5% to 7% | -6% to 8% | -11% to 17% | -18% to 26% |

[a] RI range for SEAC4RS computed from literature values as the reported median ± 1.5 * the interquartile range for Aldaif et al. (2018) and as the mean ± 2 * the standard deviation for Espinosa et al. (2018). For FIREX-AQ, the RI is assumed to be dominated by the organic aerosol component with a conservatively large range estimate for *n*, while *k* is computed following 765 Saleh et al. (2014).



**Table 5: Expected percent UHSAS sizing error associated with an atmospherically-relevant range of RIs computed for across characteristic size distributions by linearly interpolating between the laboratory response curves of NaF, (NH4)2SO4, NaCl, and SRFA as shown in Figures 5 and 6. Size distributions for SEAC4RS are from Shingler et al. (2016), while FIREX-AQ size distributions are those shown for the LAS in Figures 10 and 11.**

| Aerosol Type | RI[a] | Expected UHSAS Sizing Error (%) | | | |
|---|---|---|---|---|---|
| | | Count Mean Diameter | Volume Mean Diameter | Total Surface Area | Total Volume |
| | | | | | |
| **SEAC4RS (Aldaif et al., 2018)** | | | | | |
| Agricultural Biomass Burning | n = 1.51-1.54 | -2% to 1% | -2% to 1% | -3% to 1% | -5% to 2% |
| Wildfire Biomass Burning | n = 1.50-1.57 | -2% to 3% | -2% to 3% | -5% to 5% | -7% to 8% |
| Background | n = 1.47-1.56 | -4% to 2% | -5% to 2% | -9% to 4% | -13% to 6% |
| Biogenic | n = 1.49-1.56 | -3% to 2% | -3% to 2% | -6% to 4% | -9% to 6% |
| Free Troposphere | n = 1.46-1.54 | -5% to 1% | -5% to 1% | -10% to 1% | -15% to 2% |
| Marine | n = 1.47-1.56 | -4% to 2% | -4% to 2% | -8% to 4% | -12% to 6% |
| Urban | n = 1.48-1.58 | -4% to 3% | -4% to 3% | -7% to 7% | -11% to 11% |
| | | | | | |
| **SEAC4RS (Espinosa et al., 2018)** | | | | | |
| Biogenic | n = 1.46-1.56 k = 0.002-0.006 | -5% to 2% | -5% to 2% | -10% to 4% | -15% to 6% |
| Biomass Burning | n = 1.50-1.60 k = 0.004-0.01 | -2% to 5% | -2% to 5% | -5% to 10% | -7% to 15% |
| Urban | n = 1.47-1.57 k = 0.003-0.007 | -4% to 3% | -5% to 3% | -9% to 5% | -13% to 8% |
| | | | | | |
| **FIREX-AQ (this study)** | | | | | |
| 03 August – Upwind Leg | n = 1.47-1.56 k = 0 | -5% to 2% | -5% to 2% | -9% to 4% | -14% to 7% |
| 03 August – 0.5 hrs. Smoke age | n = 1.5-1.65 k = 0.008 | -2% to 8% | -2% to 9% | -5% to 18% | -7% to 29% |
| 03 August – 1.2 hrs. Smoke age | n = 1.5-1.65 k = 0.008 | -2% to 8% | -3% to 8% | -5% to 17% | -10% to 25% |
| 03 August – 2.4 hrs. Smoke Age | n = 1.5-1.65 k = 0.008 | -2% to 8% | -3% to 9% | -5% to 18% | -7% to 29% |
| 03 August – 3.1 hrs. Smoke Age | n = 1.5-1.65 k = 0.008 | -2% to 9% | -3% to 9% | -5% to 18% | -7% to 29% |
| 03 August – 4.3 hrs. Smoke Age | n = 1.5-1.65 k = 0.008 | -2% to 9% | -2% to 9% | -5% to 19% | -7% to 30% |
| 03 August – 4.9 hrs. Smoke Age | n = 1.5-1.65 k = 0.008 | -3% to 9% | -3% to 8% | -5% to 18% | -10% to 28% |
| 03 August – 6.1 hrs. Smoke Age | n = 1.5-1.65 k = 0.008 | -3% to 9% | -3% to 8% | -5% to 18% | -10% to 27% |
| 03 August – 7.1 hrs. Smoke Age | n = 1.5-1.65 k = 0.008 | -3% to 9% | -3% to 9% | -5% to 18% | -9% to 28% |
| 07 August – 0.9 hrs. Smoke Age | n = 1.5-1.65 k = 0.008 | -2% to 8% | -4% to 7% | -6% to 17% | -12% to 24% |
| 07 August – 2.75 hrs. Smoke Age | n = 1.5-1.65 k = 0.008 | -2% to 9% | -3% to 9% | -5% to 18% | -7% to 30% |
| 07 August – Aged Smoke | n = 1.5-1.65 k = 0.008 | -3% to 8% | -4% to 7% | -6% to 17% | -12% to 24% |

[a] RI range for SEAC4RS computed from literature values as the reported median ± 1.5 * the interquartile range for Aldaif et al.
(2018) and as the mean ± 2 * the standard deviation for Espinosa et al. (2018). For FIREX-AQ, the RI is assumed to be dominated by the organic aerosol component with a conservatively large range estimate for $n$, while $k$ is computed following Saleh et al. (2014).



**Figures:**

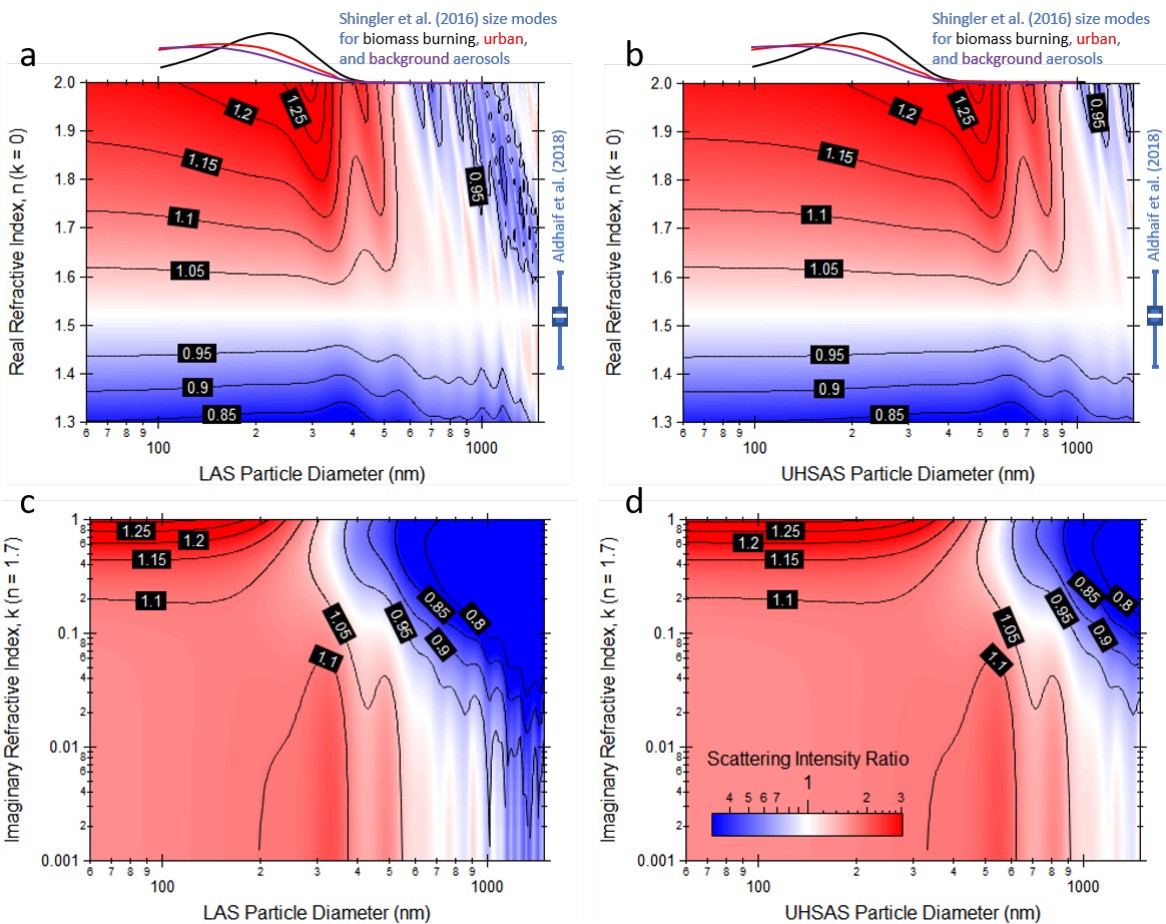

**Figure 1: Mie theory calculations of size-resolved particle scattering normalized to that for ammonium sulphate (1.52+0i) at the LAS laser wavelength (a, c) and the UHSAS laser wavelength (b, d). In panels a and b, the imaginary part of the refractive index is held**

**constant at zero, while for panels c and d, the real part of the refractive is held constant at 1.7. The coloured shading represents the ratio of the theoretical particle scattering cross section, while the contours approximate the expected sizing ratio, which is assumed to be equal to the sixth root of the scattering cross section ratio (reasonable for diameters below 300-600 nm). Average number size distribution modes for relevant atmospheric aerosol types from Shingler et al. (2016) are shown on top of panels a and b to illustrate the typical atmospheric aerosol size range, while the range of real RIs reported by Aldhaif et al. (2018) are shown to the right of these panels to**

**illustrate the typical range of atmospherically-relevant real RIs.**





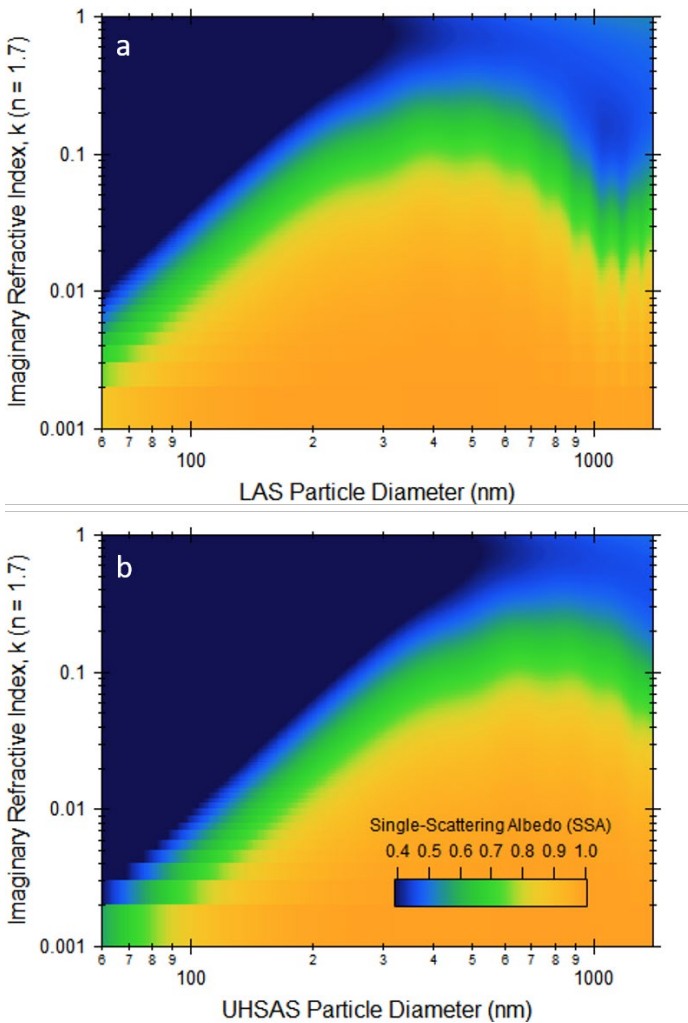

**Figure 2: Mie theory calculations of size-resolved particle single-scattering albedo for varying imaginary refractive indices at the LAS laser wavelength (a) and the UHSAS laser wavelength (b). The real part of the refractive is held constant at 1.7. Unlike Figure 1, these SSA values are computed from the total aerosol scattering and extinction coefficients that are integrated across all angles of the phase function.**



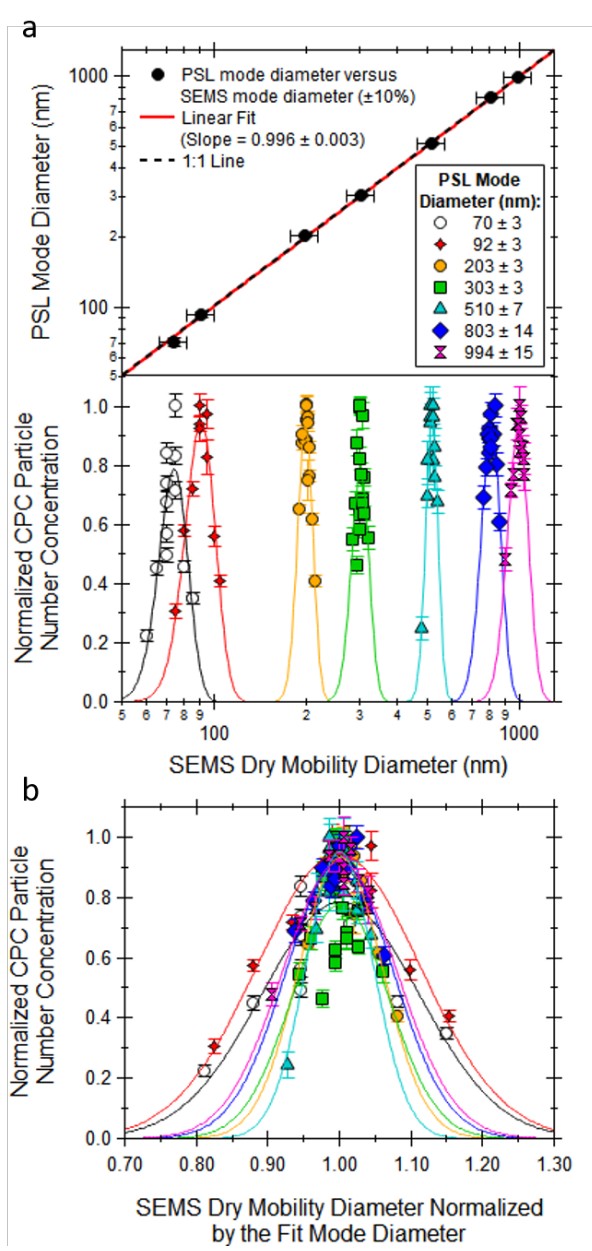

**Figure 3: SEMS DMA sizing verification with NIST-traceable polystyrene latex spheres (PSLs) size standards. Panel a compares the**
**PSL mode and SEMS fit mode mobility diameters (top) and the fits to the CPC concentration peaks (bottom) in terms of SEMS mobility**
**diameter. Panel b expresses the CPC concentration peaks in terms of the SEMS mobility diameter divided by the SEMS fit mode mobility**
**diameter.**




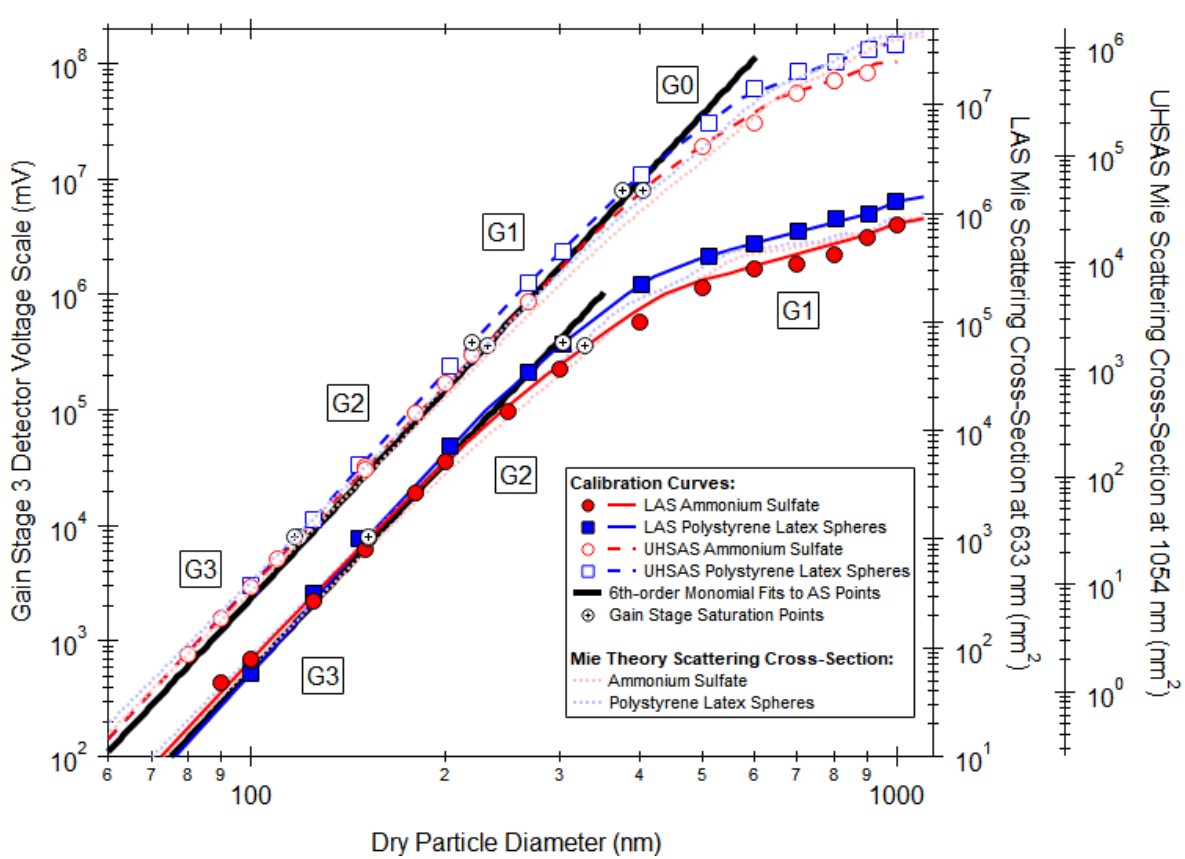

**Figure 4: Calibration curves of detector voltage versus particle size for the LAS (solid lines) and UHSAS (dashed lines) for both ammonium sulphate (red circles) and polystyrene latex spheres (blue squares). The detector responses of each gain stage are stitched together to provide a single, continuous curve across the saturation points (black, crossed circles), and the combined curve is reported in terms of the gain stage 3 voltage (left axis). Mie theory curves of particle scattering cross-section at the LAS and UHSAS laser wavelengths are shown as dotted lines for comparison with the ordinate axes aligned roughly by eye (right axes). Since the particle scattering intensity in the Rayleigh regime is expected to scale with particle size to the sixth power, solid black linear fits to the lower end of the calibration curve are included to guide the eye.**







**Figure 5: Comparison of optically-sized particle diameter as measured by the LAS (a,c,e) and the UHSAS (b,d,f) to the DMA-classified**
**mobility diameter for non-light-absorbing aerosol species listed in Table 1. Both optical sizers were calibrated to ammonium sulphate. The top two rows of panels show the ratio of the optical measurements to the mobility measurements, while the bottommost panels show the data plotted on a 1:1 line (±20% indicated).**




**Figure 6: Comparison of optically-sized particle diameter as measured by the LAS (a,c,e) and the UHSAS (b,d,f) to the DMA-classified mobility diameter for the light-absorbing aerosol species listed in Table 1. Both optical sizers were calibrated to ammonium sulphate. The top two rows of panels show the ratio of the optical measurements to the mobility measurements, while the bottommost panels show**

**the data plotted on a 1:1 line (±20% indicated).**



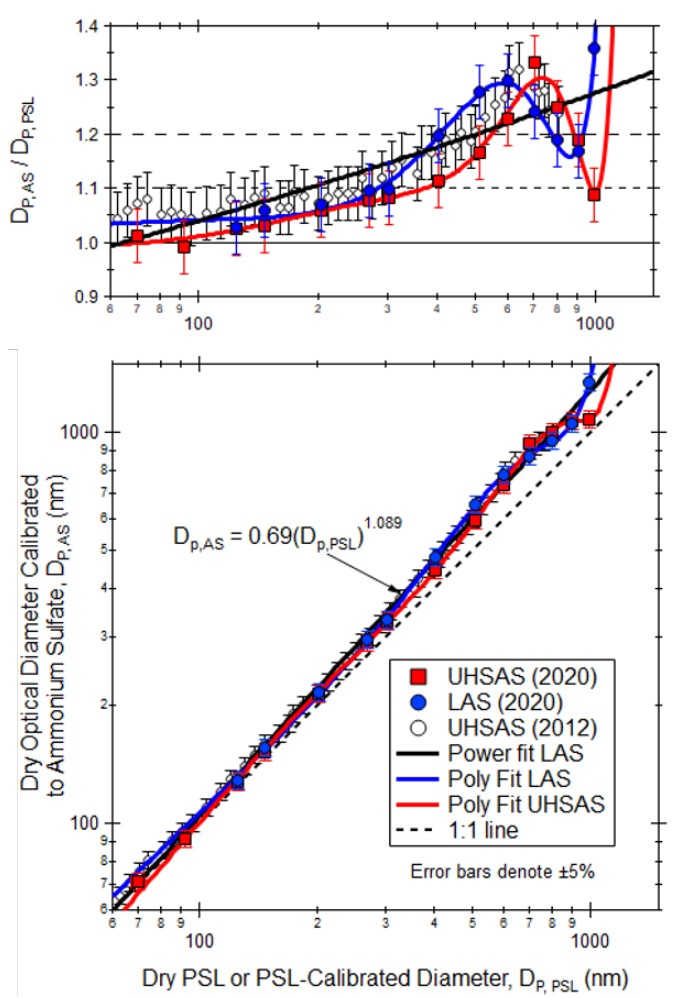

**Figure 7: Comparison of LAS and UHSAS optical diameter (calibrated using ammonium sulphate) to NIST-traceable polystyrene latex sphere (PSL) particle sizes during the 2020 laboratory experiments (red and blue points). The data are fit to a simple power law function as well as a 6ᵗʰ-order polynomial function, both with the intercept forced through the origin. Also shown for comparison as open circles are DMA-classified ammonium sulphate aerosol sizes plotted versus the UHSAS sizes during the 2012 DC3 mission (calibrated using PSLs).**




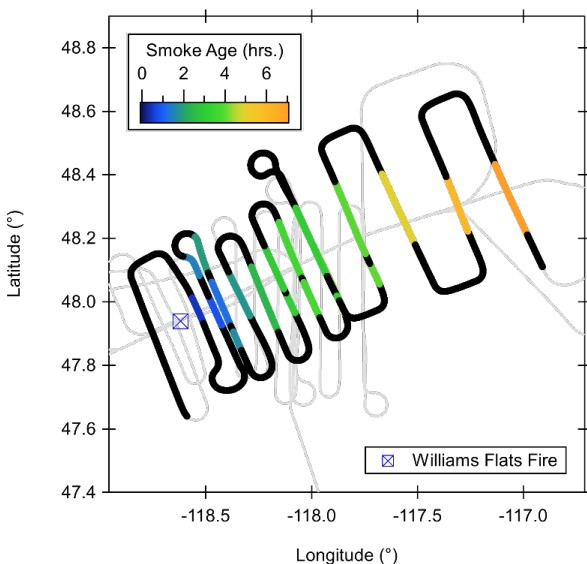

**Figure 8: NASA DC-8 flight track for the 3 August 2019 Williams Flats fire during FIREX-AQ. The thick, outlined portion of the flight track corresponds to the timeseries shown in Figure 9, while the coloured portion of the transects denotes the smoke plume extent, which is coloured by the approximate smoke age after emission. Smoke age is estimated assuming horizontal, straight-line advection between the fire and the DC-8 position at the wind speed and direction measured on the aircraft.**


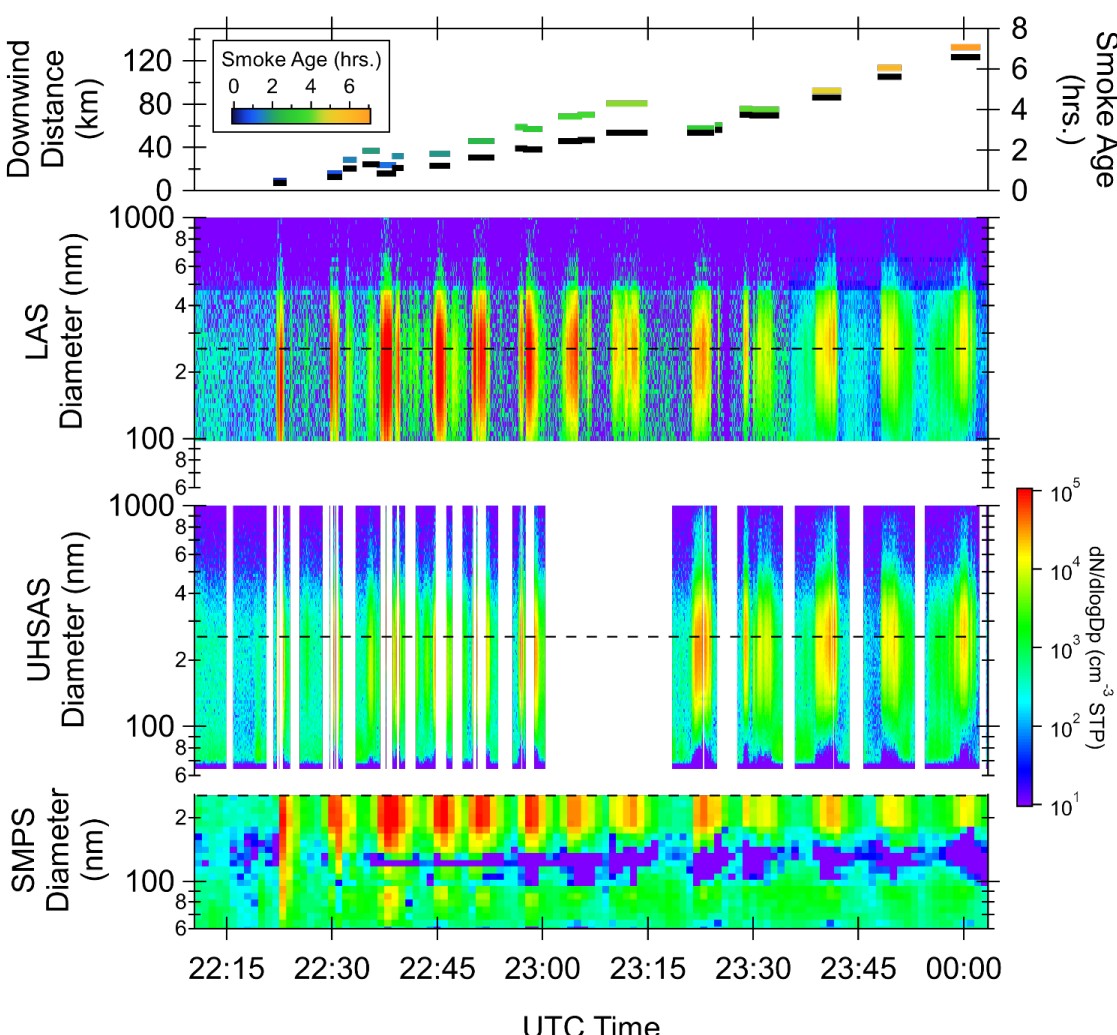


**Figure 9: Timeseries of the measured aerosol size distributions from the DC-8 on 3 August 2019 during FIREX-AQ. The top panel shows the approximate smoke age and downwind distance, corresponding to each plume transect as shown in Figure 8. The three lowest panels are the measured size distribution expressed as dN/dlogDp (cm$^{-3}$ STP). The horizontal dashed line denotes the upper size limit of the SMPS, which is plotted on a different size scale from the LAS and UHSAS.**




Figure 10: Comparison of the size distributions measured by the LAS, UHSAS, and SMPS at multiple downwind distances from the Williams Flats Fire during the 3 August 2019 FIREX-AQ flight. In some of the early plumes, the 60-second SMPS distribution was heavily skewed toward a short portion of the plume, in which case an additional LAS trace corresponding to this peak is included for the comparison. Smoke ages and downwind distances from the fire are noted for each set of size distributions and correspond to those given in Figures 8 and 9. As explained in the text, the mass to number conversion for the AMS data is very sensitive to small shifts at small sizes, hence the AMS number data below ~120 nm is fairly uncertain.



**Figure 11: Comparison of the size distributions measured by the AMS, LAS, UHSAS, and SMPS at three downwind distances from the Williams Flats Fire during the 7 August 2019 FIREX-AQ flight, including smoke sampled much farther afield. Mass distributions are computed for the LAS, UHSAS, and SMPS assuming spherical particles and using the aerosol density estimated from the mass spectrometer, while the calculation is applied in reverse to compute AMS number distributions. This density is also used to convert the AMS vacuum aerodynamic diameters to spherical-equivalent diameters.**





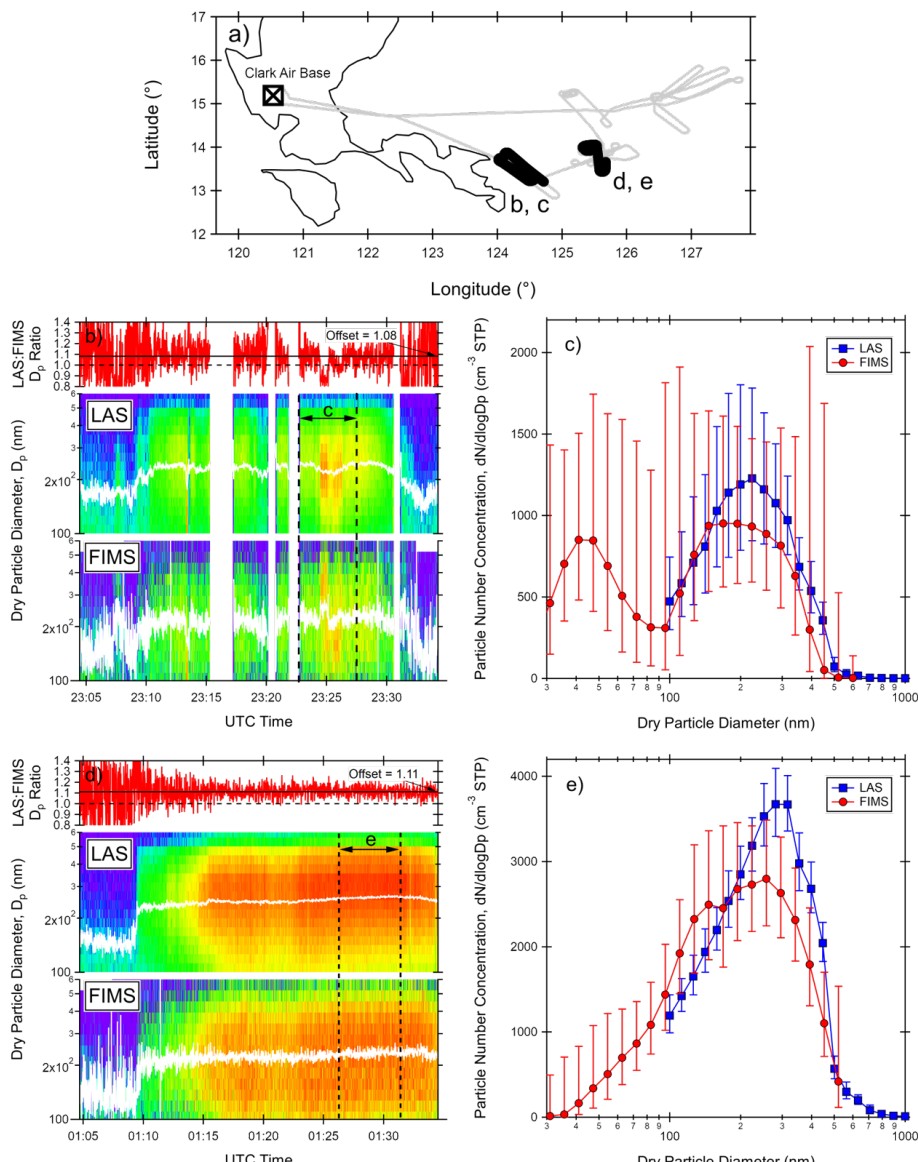

**Figure 12:** Comparison of the LAS and FIMS aerosol accumulation mode size distributions for aged biomass burning plumes encountered during CAMP²EX research flight on 16-17 September 2019. Panel a shows the flight track in gray with bolded regions indicating the plume encounters. Panels b and d show the size distribution time series for each instrument with the white trace denoting the count mean diameter (CMD) of the size distribution. Also shown is the ratio of the CMDs for the LAS and FIMS. Panels c and e are the geometric mean (✳ one geometric standard deviation) of the aerosol size distributions for level flight legs within the most intense portion of the plume intercept (denoted by the vertical dashed lines in Panels b and d).