# Peer review of "Sizing Response of the Ultra-High Sensitivity Aerosol Size Spectrometer (UHSAS) and Laser Aerosol Spectrometer (LAS) to Changes in Submicron Aerosol Composition and Refractive Index"

_Atmospheric Measurement Techniques, 2021_

## Author Response (AR1)

We thank both reviewers for their time and effort on this manuscript and for the positive and constructive comments, which we have now addressed in a revised and improved manuscript. Each reviewer's comments are given below in plain text along with the authors' response to each in bold text.

**Authors' Responses to Comments by Anonymous Referee #1**

Review of "Sizing response of the Ultra-High Sensitivity Aerosol Size Spectrometer (UHSAS) and Laser Aerosol Spectrometer (LAS) to changes in submicron aerosol composition and refractive index" by R.H. Moore, et al.

This manuscript provides experimental results and discussion regarding the sensitivity of size calibrations of two optical particle instruments to aerosol composition and refractive index. Calibration of optical particle counters to particle refractive index is extremely important and can result in significant uncertainties in sizing information if aerosol composition is not accounted for in the instrument calibration and response. This paper is very well organized, presented, and well written. The authors did a careful and detailed job of presenting laboratory experiments as well as examples from field experiments. Their results are important and provide useful context for uncertainties in aerosol size distributions for atmospherically relevant refractive indices. I have very few comments and most of them are minor. I recommend publication after addressing the comments below. Nice job!

**Thank you!  We appreciate the positive feedback.**

Line 145: First figure mentioned should be in order (figure 1).
**Done. We have removed the out of order figure references.**

Line 169: Include what Figure 1a and 1b are (Figure 1a (LAS) and Figure 1b (UHSAS)).
**Done. We added the parenthetical labels as per the reviewer.**

Line 174: Same comment as above.
**Done. We added the parenthetical labels as per the reviewer.**

Line 6: What is considered "dry" for these experiments? Was RH measured?
**We did not measure the RH during this experiment, but prior measurements with this sampling setup (~1 L min$^{-1}$ sample flow through the 1-m silica gel diffusion dryer) indicate that the RH is reduced to less than 20%.**

Line 330: How well is RH known, and is it possible that particle bound water is affecting the results?
**Based on prior work with this experimental setup, we expect that the sample stream will be dried to less than 20%RH. While we do not expect there to be condensed water that would bias the particle sizing, we speculate here that the refractive index of the salt hydrate may differ from the pure salt; however, we were unable to find literature refractive index values to either support or refute this possibility.**

Line 377: Include location of the fire (state).
**Done. We added the fire location as per the reviewer and also included a parenthetical link to its page on InciWeb.**

Tables: For tables 2-4, include/define RI "refractive index (RI)" in the caption.
**Done. We added the requested text as per the reviewer.**

Figures:
Figures 1 and 2: Please include wavelengths for the LAS and UHSAS in the captions.
**Done. We added the requested text as per the reviewer.**

Figure 5: In the caption, please include the RH of the measurements (RH<?).
**Done. Now specify that the particles were dried to less than 20%RH.**

Figure 8: Please include location of fire (state, US).
**Done. We added the fire location as per the reviewer.**

Figure 9: Please include location of fire (state, US). Were these data obtained under dry (RH<?) or ambient conditions (please state in caption).
**Done. We added the fire location and explicitly note that the size distributions are for dry particles (less than 40%RH) as per the reviewer.**

Figures 10-12: Include location of fires and whether the measurements are dry (RH<?) or ambient.
**Done. We added the fire location and explicitly note that the size distributions are for dry particles (less than 40%RH) as per the reviewer.**

References:
Check formatting, some journals are spelled out in some instances and not in others (e.g., Atmos. Phys. Chem., or Atmospheric Chemistry and Physics).
**Done. We have gone through the references to ensure that their style is consistent as per the reviewer.**

**Authors' Responses to Comments by Anonymous Referee #2**

The authors have provided a will written and structed paper to quantify the sizing errors from two commonly used optical particle sizers, the UHSAS and LAS. The fast sampling rates of these instruments make them a mainstay onboard research aircraft, however sizing errors from aerosol composition and refractive index have not been well quantified. The authors carefully presented the methodology used for the quantification of these errors through a serious of lab experiments. In addition, they provided real world examples from measurements made around wildfires. Very little work has been done to quantify the sizing error from these OPSs when measuring biomass aerosol. That makes the results presented in this publication scientifically significant.
**We thank the reviewer for the positive feedback!**

Overarching Comment
What about counting efficiency? Have any corrections for the UHSAS undercounting below 100nm been applied? Looking at Figure 10 upwind leg it appears it hasn't been applied? Is this consistent across the dataset? Please improve the UHSAS instrument description section by discussing counting efficiency and if and why corrections were or were not applied.
**We have not applied any size-dependent counting efficiency corrections to either the UHSAS or LAS laboratory data nor the airborne data, which is now noted in Sections 2.3 and 2.4. The reviewer points out a good example of this impact on the size distribution as shown in Figure 10, and we've added a sentence to the discussion in Section 3.2 highlighting this point.**

I was surprised Cai et al. (2008) wasn't mentioned anywhere in the publication.
**The important work of Cai et al. (2008) was cited and mentioned in the paper.**

Line 260, 261,386: Spelling of thermal denuders is not consistent. Line 260 there is a space between thermal and denuder. Lines 261 and 386 there is not a space.
**Done. We have modified the text per the reviewer to ensure consistency throughout the manuscript.**

Line 267: Please quantify what is considered a reasonable level? What was the range of ratios for dilutions used?
**Done. We now explicitly note $<2 \times 10^4$ cm$^{-3}$ as our rough target for the particle concentration limits, which was often driven by the TSI CPC 3010s that were also behind the dilution system. The range of dilution ratios varied from 5-20x.**

Line 278: The LAS sample flow rate is discussed however no mention of what volumetric flow rate the UHSAS was maintained at
**Done. We have added a sentence noting the UHSAS constant volumetric flow rate of 60 cm$^3$ min$^{-1}$ to the text.**

Line 302: CAMP2EX has a lower case x at the end.
**Done. We have modified the CAMP²Ex acronym to ensure consistency throughout the manuscript.**

Lines 375-380: More information about location of this fire, the altitude the measurements were made at, and average flight speed would help provide better context to the figure and the discussion that follows in 3.2. In addition, some meteorological information would be helpful. Please provide average wind speed and direction for the flight level these measurements were made.
**Done. We have added the requested information to the beginning of Section 3.2 as per the reviewer.**

Line 384: What is the local time?
**Done. We now note that local time is Pacific Daylight Time (UTC-7) as per the reviewer.**

Line 440-446: It was briefly mentioned in the instrument description section that LAS has a standard flow rate. Please acknowledge that this was accounted for and what impacts it might have on the FIMS and LAS comparison during FIREX.
**The LAS flow rate was used to convert the measured counts to concentrations, and all comparisons in this paper are made in terms of particle concentration, which places all instruments on an equal footing. Consequently, we would not expect there to be an impact on the FIMS-LAS comparison during CAMP²Ex or on the multi-instrument comparison during FIREX-AQ.**

**Controlling the LAS flow to a mass flow instead of a volumetric flow may change the particle velocity through the laser as well as the sheath:aerosol flow ratio (which may influence the shape of the particle beam). None of these potential impacts appear to affect the particle size for the laboratory and low-level plume sampling in this study; however, the effects may be more significant for higher altitude measurements. We hope to explore some of these pressure-dependence instrument performance characteristics in a follow-on paper.**

Lines 825-835: Figure 5 and 6 could be better organized and annotated. a and b appear to be zoomed in views of c and d? I feel it would be more logical if the wide view is shown first as a and b and this was discussed in the captions. In addition, these are organized as columns. I recommend labeling the top of the left hand column as LAS and the right hand column as UHSAS instead of repeating it on every image.
**Done. We have modified the figure as per the reviewer.**

Line 850: Figure 8. Can more information be provided on the location of this fire? Perhaps a map indicating the location of the fire would accomplish this.
**Done. Have added a link to the fire InciWeb page to both the figure caption and the text.**

Line 855: Figure 9. Please indicate what Local time is for these measurements
**Done. We now note that local time is Pacific Daylight Time (UTC-7) in the caption as per the reviewer.**

---

## Author Response (AR2)

**We thank the editor for the thorough re-read of the manuscript and apologize for missing these corrections in our last revision. All corrections are now implemented in the revised manuscript.**

Very nice job on a well-written paper, and on addressing the reviewers' suggested minor corrections. However, a couple of the reviewers' corrections were not fully addressed:

Reviewer 1:

-Requested that Figs. 10-12 specify the RH of the measurements. This was done for Figs 10 and 11, but not for Fig 12. Please specify the RH for Fig. 12

**Done. We now note in the caption of Figure 12 that the RH < 30%.**

Reviewer 2:

-Lines 375-380 requested that the altitude of the measurements, and the average wind speed and direction at flight level be provided. I do not see any mention of that at the beginning of Section 3.2. Please add in that requested information.

**Done. We now note on Line 381 that the altitude was ~ 2700 meters and on Line 383 that winds were approximately 5 m/s from the west southwest.**

-Requested that Figs 5 and 6 be reformatted with a and b being the wide view and that LAS and UHSAS be labeled at the top of the figures. These changes were made for Fig 5, but not for Fig 6. Please reformat Fig. 6 to match Fig. 5

**Done. We have replaced Figure 6 with the correct, updated version.**